# Hierarchical Refinement: Optimal Transport to Infinity and Beyond

Peter Halmos [1] [*]   Julian Gold [1] [2] [*]   Xinhao Liu [1]   Benjamin J. Raphael [1]

## Abstract

Optimal transport (OT) has enjoyed great success in machine learning as a principled way to align datasets via a least-cost correspondence, driven in large part by the runtime efficiency of the Sinkhorn algorithm (Cuturi, 2013). However, Sinkhorn has quadratic space and time complexity in the number of points, limiting scalability to larger datasets. Low-rank OT achieves linear complexity, but by definition, cannot compute a one-to-one correspondence between points. When the optimal transport problem is an assignment problem between datasets then an optimal mapping, known as the *Monge map*, is guaranteed to be a bijection. In this setting, we show that the factors of an optimal low-rank coupling co-cluster each point with its image under the Monge map. We leverage this invariant to derive an algorithm, *Hierarchical Refinement* (`HiRef`), that dynamically constructs a multiscale partition of each dataset using low-rank OT subproblems, culminating in the bijective Monge map. Hierarchical Refinement runs in log-linear time and linear space, retaining the advantages of low-rank OT while overcoming its limited resolution. We demonstrate the advantages of Hierarchical Refinement on several datasets, including ones containing over a million points, scaling full-rank OT to problems previously beyond Sinkhorn's reach.

## 1. Introduction

Optimal transport (OT) is a mathematical framework for comparing probability distributions $\mu$ and $\nu$. Given a cost function $c$, the *Monge problem* is to find a mapping $T$ transforming a distribution $\mu$ into $\nu$ (i.e. $T_\sharp \mu = \nu$) with least-cost. A relaxation of this problem, called the *Kantorovich prob-*

*lem*, instead seeks a least-cost coupling $\gamma$ between $\mu$ and $\nu$. In the Kantorovich formulation, mass splitting is allowed and thus a solution always exists; in contrast, a Monge map between $\mu$ and $\nu$ may not exist. When a Monge map $T$ does exist, the solution to the Kantorovich problem is a coupling $\gamma = (\mathrm{id} \times T)_\sharp \mu$ supported on its graph, and the Monge and Kantorovich problems coincide (Brenier, 1991).

When $\mu$ and $\nu$ are discrete uniform measures on $n$ points the optimal transport problem reduces to an assignment problem. Classical algorithms such as the Hungarian algorithm and Network Simplex (Tarjan, 1997; Orlin, 1997), solve this in cubic time. The Sinkhorn algorithm (Cuturi, 2013) solves the entropy-regularized Kantorovich problem with quadratic runtime, greatly expanding the applicability of computational OT. However, the Sinkhorn algorihtm requires quadratic space to store the coupling $\gamma$.

In recent years, OT has found numerous applications in machine learning and across science, including: domain adaptation (Courty et al., 2014; Solomon et al., 2015), self-attention (Tay et al., 2020; Sander et al., 2022; Geshkovski et al., 2023), computational biology (Schiebinger et al., 2019; Yang et al., 2020; Zeira et al., 2022; Bunne et al., 2023; Halmos et al., 2025b; Klein et al., 2025), unpaired data translation (Korotin et al., 2021; De Bortoli et al., 2024; Tong et al., 2024; Klein et al., 2024), and alignment problems in transformers and large language models (Melnyk et al., 2024; Li et al., 2024). The *least-cost* principle of optimal transport is crucial for training high-quality generative models using Schrödinger bridges, flow-matching, diffusion models, or neural ordinary differential equations (Finlay et al., 2020; Tong et al., 2024; De Bortoli et al., 2024; Kornilov et al., 2024; Klein et al., 2024). These models typically require millions to hundreds of millions of data-points to achieve high-performance at scale (Ramesh et al., 2021), limiting the scope of OT for generative modeling.

As modern datasets grow to have tens of thousands or even millions of points, the quadratic space and time complexity of Sinkhorn becomes increasingly prohibitive. This limitation is widely recognized in the machine learning literature, with (De Bortoli et al., 2024) noting that the quadratic complexity of optimal transport renders its application to modern datasets on the order of millions of points impractical. A number of approaches have been proposed to address scal-

---

[*]Equal contribution [1]Department of Computer Science, Princeton University [2]Center for Statistics and Machine Learning, Princeton University. Correspondence to: Benjamin J. Raphael <braphael@princeton.edu>.

*Proceedings of the 42nd International Conference on Machine Learning*, Vancouver, Canada. PMLR 267, 2025. Copyright 2025 by the author(s).

ing OT to massive datasets which avoid instantiating a full coupling matrix. Mini-batch OT (Genevay et al., 2018) improves scalability, but incurs significant biases (Sommerfeld et al., 2019; Korotin et al., 2021; Fatras et al., 2021a) as each mini-batch alignment is a poor representation of the global coupling. Multiple works have investigated the theoretical properties of mini-batch estimators of the coupling (Fatras et al., 2020; 2021b), while others have attempted to mitigate this bias using partial or unbalanced OT that allows mass variation between mini-batches (Nguyen et al., 2022a; Fatras et al., 2021a). However, these approaches introduce additional hyperparameters to control the degree of unbalancedness, and ultimately remain biased, local approximations of the global coupling.

Neural optimal transport methods (Makkuva et al., 2020; Bunne et al., 2023; Fan et al., 2023; Korotin et al., 2023; Buzun et al., 2024), parametrize the Monge map as a neural network instead of materializing a quadratic coupling matrix. However, these methods have noted limitations recovering faithful maps (Korotin et al., 2021).

Another approach to improve space complexity of OT is to introduce a *low-rank* constraint on the coupling matrix in the Kantorovich problem. This has been done by parameterizing the coupling through a set of low-rank factors (Scetbon et al., 2021; 2022; Scetbon & Cuturi, 2022; Scetbon et al., 2023; Halmos et al., 2024) or by using a proxy objective for the low-rank problem, factoring the transport through a small number of anchor points (Forrow et al., 2019; Lin et al., 2021). For a given rank $r$ these approaches have $O(nr)$ space complexity, enabling *linear* time and space scaling. Low-rank OT has been used successfully on datasets on the order of $10^5$ samples with ranks on the order of $10^1$ (Scetbon et al., 2023; Halmos et al., 2024; 2025a; Klein et al., 2025), but computing *full-rank* couplings between datasets of sizes on the order of $10^5$ and greater has not yet been accomplished.

**Contributions**  We introduce Hierarchical Refinement (`HiRef`), an algorithm to scalably compute a full-rank alignment between two equally-sized input datasets X and Y by solving a hierarchy of low-rank OT sub-problems. The success of this refinement is driven by a theoretical result, Proposition 3.1, stating that factors of an optimal low-rank coupling between X and Y co-cluster points X with their image under the Monge map. We use Proposition 3.1 recursively to obtain increasingly fine partitions of X and Y. At each scale, the solutions to low-rank OT sub-problems are bijections between the partitions of X and Y. Iterating to the finest scale gives a bijection between X and Y.

Hierarchical Refinement constructs a *multiscale partition* of each dataset, and thus is related to (Gerber & Maggioni, 2017), which introduced a general framework for multiscale optimal transport using such partitions, and the earlier work

of (Mérigot, 2011). Unlike (Mérigot, 2011; Gerber & Maggioni, 2017), Hierarchical Refinement (i) does not assume multiscale partitions for each dataset are given, instead constructing them on the fly; and (ii) operates intrinsically to the data, without a mesh or anchor points in the ambient space of the data, avoiding the curse of dimensionality.

We demonstrate that Hierarchical Refinement computes OT maps efficiently in high-dimensional spaces, often matching or even outperforming Sinkhorn in terms of primal cost. Moreover, `HiRef` has linear space complexity and time complexity scaling log-linearly in the dataset size. Unlike low-rank OT, Hierarchical Refinement places X and Y in bijective correspondence. Hierarchical Refinement scales to over a million points, enabling the use of OT on massive datasets without incurring the bias of mini-batching.

## 2. Background and Related Work

Suppose $\mathsf{X} = \{\mathbf{x}_i\}_{i=1}^n$ and $\mathsf{Y} = \{\mathbf{y}_j\}_{j=1}^m$ are datasets in the same metric space $(\mathcal{X}, \mathsf{d}_\mathcal{X})$. Let $c : \mathcal{X} \times \mathcal{X} \to \mathbb{R}_+$ be a cost function. This cost $c$ is often assumed to satisfy strict convexity or to be a metric. Datasets X and Y are represented as discretely supported probability measures $\mu = \sum_{i=1}^n \mathbf{a}_i \delta_{\mathbf{x}_i}$ and $\nu = \sum_{j=1}^m \mathbf{b}_j \delta_{\mathbf{y}_j}$ for probability vectors $\mathbf{a} \in \Delta_n$ and $\mathbf{b} \in \Delta_m$. Throughout, $\Delta_k$ denotes the *$k$-simplex* $\{\mathbf{p} \in \mathbb{R}_+^k : \sum_i \mathbf{p}_i = 1\}$, the set of probability vectors of length $k$.

**Monge Problem**  Optimal transport has its origin in the *Monge problem* (Monge, 1781), concerned with finding an optimal map $T : \mathsf{X} \to \mathsf{Y}$ pushing $\mu$ forward to $\nu$:

$$\mathrm{M}_c(\mu, \nu) = \min_{T : T_\sharp \mu = \nu} \mathbb{E}_\mu c(x, T(x)) . \quad (1)$$

Above, $T_\sharp \mu$ is the pushforward of $\mu$ under $T$, the measure on Y with $T_\sharp \mu(B) := \mu(T^{-1}(B))$ for any (measurable) set $B \subset \mathsf{Y}$. In general, a Monge map may not exist (e.g. if $m > n$). However, when $|\mathsf{X}| = |\mathsf{Y}| = n$ and $\mathbf{a}, \mathbf{b}$ are uniform then the Monge problem becomes the *assignment problem* and has a bijective solution (Thorpe, 2018).

**Kantorovich Problem**  The *Kantorovich problem* (Kantorovich, 1942) was introduced as a relaxation of the Monge problem. In contrast to the Monge problem, the Kantorovich problem allows mass-splitting and a solution is always guaranteed to exist. Define the *transport polytope* $\Pi_{\mathbf{a},\mathbf{b}}$ as the following set of coupling matrices

$$\Pi_{\mathbf{a},\mathbf{b}} := \left\{ \mathbf{P} \in \mathbb{R}_+^{n \times m} : \mathbf{P}\mathbf{1}_m = \mathbf{a}, \mathbf{P}^\top \mathbf{1}_n = \mathbf{b} \right\}, \quad (2)$$

respectively with left (or "source") marginal $\mathbf{a}$ and with right (or "target") marginal $\mathbf{b}$. For the cost $c(\cdot, \cdot)$, define the cost matrix $\mathbf{C}$ by $\mathbf{C}_{ij} = c(x_i, y_j)$. In this discrete setting, the Kantorovich problem seeks a least cost coupling matrix

$\mathbf{P} \in \Pi_{\mathbf{a},\mathbf{b}}$ between the probability vectors $\mathbf{a}, \mathbf{b}$ associated to each measure $\mu, \nu$:

$$W_c(\mu, \nu) = \min_{\mathbf{P} \in \Pi_{\mathbf{a},\mathbf{b}}} \langle \mathbf{C}, \mathbf{P} \rangle_F . \tag{3}$$

The optimal value $W_c(\mu, \nu)$ of (3) is called the *c-Wasserstein distance* between $\mu$ and $\nu$.

**Sinkhorn Algorithm and the $\epsilon$-schedule**   The Sinkhorn algorithm (Cuturi, 2013) relaxes the classical linear-programming formulation of optimal transport by solving an entropy regularized version of (3),

$$W_\epsilon(\mu, \nu) := \min_{\mathbf{P} \in \Pi_{\mathbf{a},\mathbf{b}}} \langle \mathbf{C}, \mathbf{P} \rangle_F - \epsilon H(\mathbf{P}), \tag{4}$$

where $H(\mathbf{P}) := -\sum_{ij} \mathbf{P}_{ij}(\log \mathbf{P}_{ij} - 1)$ is the Shannon entropy, and the parameter $\epsilon > 0$ is the regularization strength. The Sinkhorn algorithm improved the $O(n^3 \log n)$ time complexity of classical techniques used for OT such as the Hungarian algorithm (Kuhn, 1955) and Network Simplex (Orlin, 1997; Tarjan, 1997) to $O(n^2 \log n)$ (Luo et al., 2023). As $\epsilon \downarrow 0$, the optimal coupling $\mathbf{P}^{\star,\epsilon}$ for (4) converges to a sparse optimal coupling for (3) at an extremal point of the transport polytope (c.f. (Peyré & Cuturi, 2019)). However, the number of iterations required scales as $\mathrm{poly}(1/\epsilon)$, diverging as $\epsilon$ decreases. A technique used to improve this scaling is the $\epsilon$-schedule, an adaptive, monotone-decreasing and step-dependent set of entropy parameters $\epsilon_1 > \epsilon_2 > \cdots > \epsilon_{t_{\mathrm{fin}}}$. This anneals Problem 4 from high-entropy to low-entropy, gradually driving a dense initial condition to a sparse solution with a $\log(1/\epsilon)$ rate (Chen et al., 2023).

**Low-rank Optimal Transport**   The nonnegative rank $\mathrm{rk}_+(\mathbf{M})$ of a nonnegative matrix $\mathbf{M} \succcurlyeq 0$ is the smallest number of nonnegative rank-1 matrices summing to $\mathbf{M}$; i.e. $\mathrm{rk}_+(\mathbf{M})$ is the smallest integer $z$ such that there exist nonnegative vectors $\mathbf{q}_1, \ldots, \mathbf{q}_z \succcurlyeq 0$ and $\mathbf{r}_1, \ldots, \mathbf{r}_z \succcurlyeq 0$ satisfying $\mathbf{M} = \sum_{i=1}^z \mathbf{q}_i \mathbf{r}_i^\top$. Let $\Pi_{\mathbf{a},\mathbf{b}}(r) := \{\mathbf{P} \in \Pi_{\mathbf{a},\mathbf{b}} : \mathrm{rk}_+(\mathbf{P}) = r\}$ be the set of rank-$r$ couplings. The low-rank Wasserstein problem for general cost matrix $\mathbf{C}$ is:

$$\mathbf{P}^\star = \arg\min_{\mathbf{P} \in \Pi_{\mathbf{a},\mathbf{b}}(r)} \langle \mathbf{C}, \mathbf{P} \rangle_F . \tag{5}$$

From (Cohen & Rothblum, 1993), each $\mathbf{P} \in \Pi_{\mathbf{a},\mathbf{b}}(r)$ may be decomposed as

$$\mathbf{P} = \sum_{i=1}^r (1/\mathbf{g}_i)\mathbf{Q}_{\cdot,i}\mathbf{R}_{\cdot,i}^\top := \mathbf{Q}\mathrm{diag}(1/\mathbf{g})\mathbf{R}^\top, \tag{6}$$

where $\mathbf{g} \in \Delta_r, \mathbf{Q} \in \Pi_{\mathbf{a},\mathbf{g}}$ and $\mathbf{R} \in \Pi_{\mathbf{b},\mathbf{g}}$. This factorization was introduced to optimal transport by (Scetbon et al., 2021) in the context of the general low-rank problem (5). The factors $\mathbf{Q}$ and $\mathbf{R}$ constitute co-clusterings of datasets $\mathsf{X}$ and $\mathsf{Y}$ onto the *same* set of $r$ components. Other factorizations have recently been proposed (Halmos et al., 2024), using $\mathbf{Q}, \mathbf{R}$ and an intermediate latent coupling $\mathbf{T}$ to solve (5) where $\mathsf{X}$ and $\mathsf{Y}$ have $r_1$ and $r_2$ components, respectively.

**Hierarchical and Multiscale Approaches to OT**   Hierarchical optimal transport (Schmitzer & Schnörr, 2013) is a variant of OT modeling data and transport at two scales, using Wasserstein distances as the coarse-scale ground costs. It has been applied to document representation (Yurochkin et al., 2019), domain adaptation (El Hamri et al., 2022), sliced Wasserstein distances (Bonneel et al., 2015; Nguyen et al., 2022b) and to give a discrete formulation of transport between Gaussian mixture models (Chen et al., 2018; Delon & Desolneux, 2020). These works build interpretable, coarse-grained structure into a single coupling, rather than solving for a sequence of couplings at progressively finer scales as in the present work.

Multiscale approaches to OT generalize hierarchical OT to a progression of scales. Building on the semidiscrete approach of (Aurenhammer et al., 1998), (Mérigot, 2011) uses Lloyd's algorithm to progressively coarse-grain the target measure. More recently, using a regular family of multiscale partitions (Definition C.3) on each dataset, (Gerber & Maggioni, 2017) formalize a general hierarchical approach to the Kantorovich problem (3). They propose: (i) solving a Kantorovich problem between the coarsest partitions of $\mathsf{X}$ and $\mathsf{Y}$ in their respective multiscale families; and (ii) propagation of the optimal coupling at scale $t \in \{1, \ldots, \kappa - 1\}$ to initialize the optimization at scale $t + 1$. They take as input a chain of partitions and measures across scales $(\mathsf{X}^{(1)}, \mu_1) \to \cdots \to (\mathsf{X}^{(\kappa)}, \mu_\kappa)$ and $(\mathsf{Y}^{(1)}, \nu_1) \to \cdots \to (\mathsf{Y}^{(\kappa)}, \nu_\kappa)$ where each dataset $\mathsf{X}, \mathsf{Y}$ is identified with the trivial partitions $\mathsf{X}^{(\kappa)} = \{\{\mathbf{x}\} : \mathbf{x} \in \mathsf{X}\}$ and $\mathsf{Y}^{(\kappa)} = \{\{\mathbf{y}\} : \mathbf{y} \in \mathsf{Y}\}$. At the finest scale $\kappa$, (Gerber & Maggioni, 2017) recover the original datasets and a near optimal coupling for (3).

A naive implementation of the above idea requires quadratic memory complexity, but (Gerber & Maggioni, 2017) propose several propagation strategies to mitigate this, following (Glimm & Henscheid, 2013; Oberman & Ruan, 2015; Schmitzer, 2016). These strategies use the optimal coupling at scale $t$ to restrict the support of the coupling computed at the next scale using local optimality criteria. In the next section, we give our own such criterion, Proposition 3.1.

# 3. Hierarchical Refinement

## 3.1. Low-rank optimal transport co-clusters source-target pairs under the Monge map

We first show that under a few assumptions, the optimal low-rank factors $(\mathbf{Q}^\star, \mathbf{R}^\star)$ for a *variant* of the low-rank Wasserstein problem (5) have qualities suited to our refinement strategy. Specifically, we parameterize low-rank couplings $\mathbf{P}$ of rank-$r$ using the factorization $\mathbf{P} = \mathbf{Q}\mathrm{diag}(1/\mathbf{g})\mathbf{R}^\top$ of (Scetbon et al., 2021), fixing $\mathbf{g} \in \Delta_r$ to be uniform. Define

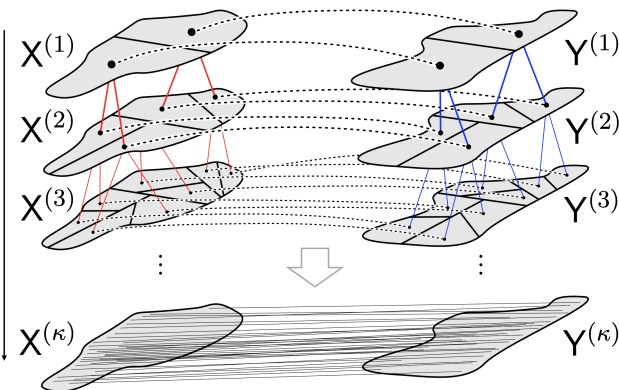

*Figure 1.* Hierarchical Refinement algorithm: low-rank optimal transport is used to progressively refine partitions at the previous scale, with the coarsest scale partitions denoted $X^{(1)}, Y^{(1)}$, and the finest scale partitions $X^{(\kappa)}, Y^{(\kappa)}$ corresponding to the individual points in the datasets.

the following variant of (5):

$$(\mathbf{Q}^\star, \mathbf{R}^\star) = \arg \min_{(\mathbf{Q},\mathbf{R})} \left\langle \mathbf{C}, \mathbf{Q}\mathrm{diag}(1/\mathbf{g})\mathbf{R}^\top \right\rangle_F \quad (7)$$

$$\text{s.t.} \quad \mathbf{Q} \in \Pi_{\mathbf{a},\mathbf{g}}, \ \mathbf{R} \in \Pi_{\mathbf{b},\mathbf{g}}, \ \mathbf{g} = (1/r)\mathbf{1}_r$$

Proposition 3.1 below is the main structural result behind Hierarchical Refinement. It says that when optimal $\mathbf{Q}^\star$ and $\mathbf{R}^\star$ for (7) correspond to hard-clusterings (partitions) of each dataset, given by clustering functions $\mathsf{q}^\star : \mathsf{X} \to [r], \mathsf{r}^\star : \mathsf{Y} \to [r]$, one has $\mathsf{q}^\star = \mathsf{r}^\star \circ T^\star$, where $T^\star$ is a Monge map.

**Proposition 3.1** (Optimal low-rank factors co-cluster Monge pairs)**.** *Let* $\mathsf{X}, \mathsf{Y} \subset \mathbb{R}^d$ *with* $|\mathsf{X}| = |\mathsf{Y}| = n$, *with cost matrix* $\mathbf{C}$ *that is strictly $r$-Monge separable (Definition B.2). Let* $\mathbf{a}, \mathbf{b} \in \Delta_n$ *be uniform so that a Monge map* $T^\star : \mathsf{X} \to \mathsf{Y}$ *exists. If* $(\mathbf{Q}^\star, \mathbf{R}^\star)$ *are minimizers of (7) and correspond to clustering functions* $\mathsf{q}^\star : \mathsf{X} \to [r], \mathsf{r}^\star : \mathsf{Y} \to [r]$, *then for all* $\mathbf{x} \in \mathsf{X}$ *one has* $\mathsf{q}^\star(\mathbf{x}) = \mathsf{r}^\star(T^\star(\mathbf{x}))$.

The proof of Proposition 3.1 is in two steps. First, we use the existence of a Monge map and its coupling $\mathbf{P}^\dagger$ to permute the cost $\mathbf{C}$ to cost $\mathbf{C}^\dagger$ (Definition B.1) for which the identity matrix is a Monge map. Second, supposing that strict $r$-Monge separability (Definition B.2) holds, we show the solution to Problem 7 with cost $\mathbf{C}^\dagger$ is symmetric, so that $\min_{\mathbf{Q},\mathbf{R} \in \Pi_{\mathbf{a},\mathbf{g}}} \langle \mathbf{C}^\dagger, \mathbf{Q}\mathbf{R}^\top \rangle_F = \min_{\mathbf{Q} \in \Pi_{\mathbf{a},\mathbf{g}}} \langle \mathbf{C}^\dagger, \mathbf{Q}\mathbf{Q}^\top \rangle_F$. Returning to the coordinate frame of the original cost $\mathbf{C}$, we find that $\mathbf{Q} = \mathbf{P}^\dagger \mathbf{R}$, implying Proposition 3.1. We note that when $r = 2$, optimal $\mathbf{Q}, \mathbf{R}$ are hard-partitions (Lemma B.5) automatically satisfying one of the assumptions of Proposition 3.1.

### 3.2. Hierarchical Refinement Algorithm

The Hierarchical Refinement algorithm (Algorithm 1) uses Proposition 3.1 to guarantee that each low-rank step co-

clusters the datasets optimally, in that $\mathbf{x}$ and $T^\star(\mathbf{x})$ are assigned the same label by $\mathsf{q}^\star$ and $\mathsf{r}^\star$. Using the same label set to partition $\mathsf{X}$ and $\mathsf{Y}$ automatically places the blocks of each partition in bijective correspondence. One then recurses on each pair of corresponding blocks (which we call a *co-cluster*) at the previous scale, until all blocks have size one. This guarantee holds despite that optimal $(\mathbf{Q}^\star, \mathbf{R}^\star)$ for (7) may not constitute an optimal triple $(\mathbf{Q}^\star, \mathbf{R}^\star, \mathbf{g}^\star)$ for the original low-rank problem (5) under the (Scetbon et al., 2021) factorization.

A hierarchy-depth $\kappa$ denotes the total number of times Algorithm 1 refines the initial trivial partitions $\{\mathsf{X}\}, \{\mathsf{Y}\}$. The effective rank at scale $t$ is $\rho_t := \prod_{s=1}^t r_s$, given rank-annealing schedule $(r_1, r_2, \ldots, r_\kappa)$ for which $\rho_\kappa$ divides $n$. The base rank is $r_{\text{base}} = \frac{n}{\rho_\kappa}$. Note that $n/\rho_t$ is also the size of each partition at scale $t$: $n/\rho_t = |\mathsf{X}^{(t)}| = |\mathsf{Y}^{(t)}|$, and that any sequence of any factorization of $n$ corresponds to a rank-annealing schedule.

**Proposition 3.2.** *For any $n$, there exists a rank-schedule $(r_1, \cdots, r_\kappa)$ factorizing $n$ such that all partitions of Algorithm 1 at level $t \in [0 : \kappa - 1]$ satisfy strict $r_{t+1}$-Monge separability (Definition B.2). Let* LROT *denote an optimal rank-$r$ solver for (7) over hard-partitions. For any satisfying rank-schedule, the map returned by Algorithm 1 is optimal and supported on the graph of the Monge map $T^\star$.*

*Proof.* Existence follows from the trivial $(r_1) = (n)$ rank-schedule. For any schedule $(r_1, \cdots, r_\kappa)$ satisfying Monge separability, applying the invariant of Proposition 3.1 inductively on $t$ to level $\kappa$ yields $n$ tuples $\{(\mathbf{x}, T^\star(\mathbf{x}))\}$ containing each $\mathbf{x} \in \mathsf{X}$ and its image $T^\star(\mathbf{x})$ under the Monge map. $\square$

If the black-box subroutine LROT in Algorithm 1 solves (7) optimally, then Hierarchical Refinement is guaranteed to recover a Monge map. In practice, we implement LROT using the low-rank solver (Halmos et al., 2024) and enforce that inner marginal $\mathbf{g}$ is uniform.

Let $\Gamma_{t,q}$ denote the $q$-th co-cluster at scale $t$ generated by Hierarchical Refinement:

$$\Gamma_{t,q} := \left\{ (\mathbf{x}, \mathbf{y}) : \mathbf{x} \in \mathsf{X}_q^{(t)}, \ \mathbf{y} \in \mathsf{Y}_q^{(t)} \right\}, \quad (8)$$

where $\mathsf{X}^{(t)} = \{\mathsf{X}_q^{(t)}\}_{q=1}^{\rho_t}, \mathsf{Y}^{(t)} = \{\mathsf{Y}_q^{(t)}\}_{q=1}^{\rho_t}$, and define the co-clustering $\Gamma_t$ at scale $t$ by:

$$\Gamma_t := \left\{ (\mathsf{X}_q^{(t)}, \mathsf{Y}_q^{(t)}) \right\}_{q=1}^{\rho_t}.$$

At scale $t \in [\kappa]$, Hierarchical Refinement refines $\Gamma_t$ to $\Gamma_{t+1}$ by running a rank $r_{t+1}$ low-rank optimal transport problem between uniform $\mathbf{g}_{t+1} = (1/r_{t+1})\mathbf{1}_{r_{t+1}}$ and measures supported on each pair $(\mathsf{X}_q^{(t)}, \mathsf{Y}_q^{(t)})$ in $\Gamma_t$ for $q \in [\rho_t]$, yielding

**Algorithm 1** Hierarchical Refinement

**Require: Data** $\mathsf{X}, \mathsf{Y}$; **Low-rank OT solver** $\mathrm{LROT}(\cdot)$;
   **Rank schedule** $(r_1, r_2, \ldots, r_\kappa)$; **Base rank** $r_{\mathrm{base}}$ (=1).
   **Initialize:**

1: $t \leftarrow 0, \Gamma_0 \leftarrow \{ (\mathsf{X}, \mathsf{Y}) \}$
2: **while** $\exists (\mathsf{X}_q^{(t)}, \mathsf{Y}_q^{(t)}) \in \Gamma_t$ **such that**
3: $\qquad \qquad \min\{|\mathsf{X}_q^{(t)}|, |\mathsf{Y}_q^{(t)}|\} > r_{\mathrm{base}}$ **do**
4: $\quad \Gamma_{t+1} \leftarrow \varnothing$
5: $\quad$ **for** $(\mathsf{X}_q^{(t)}, \mathsf{Y}_q^{(t)}) \in \Gamma_t$ **do**
6: $\qquad$ **if** $\min\{|\mathsf{X}_q^{(t)}|, |\mathsf{Y}_q^{(t)}|\} \leq r_{\mathrm{base}}$ **then**
7: $\qquad \quad \Gamma_{t+1} \leftarrow \Gamma_{t+1} \cup \{(\mathsf{X}_q^{(t)}, \mathsf{Y}_q^{(t)})\}$
8: $\qquad$ **else**
9: $\qquad \quad \mu_{\mathsf{X}_q^{(t)}} = \frac{1}{|\mathsf{X}_q^{(t)}|} \sum_{\mathbf{x} \in \mathsf{X}_q^{(t)}} \delta_{\mathbf{x}}$
10: $\qquad \quad \mu_{\mathsf{Y}_q^{(t)}} = \frac{1}{|\mathsf{Y}_q^{(t)}|} \sum_{\mathbf{y} \in \mathsf{Y}_q^{(t)}} \delta_{\mathbf{y}}.$
11: $\qquad \quad \mathbf{g}_{t+1} \leftarrow (1/r_{t+1}) \mathbf{1}_{r_{t+1}}$
12: $\qquad \quad (\mathbf{Q}, \mathbf{R}) \leftarrow \mathrm{LROT}(\mu_{\mathsf{X}_q^{(t)}}, \mu_{\mathsf{Y}_q^{(t)}}, \mathbf{g}_{t+1})$
13: $\qquad \quad$ **for** $z = 1 \to r_{t+1}$ **do**
14: $\qquad \qquad \mathsf{X}_z^{(t+1)} \leftarrow \mathrm{Assign}(\mathsf{X}^{(t)}, \mathbf{Q}, z)$
15: $\qquad \qquad \mathsf{Y}_z^{(t+1)} \leftarrow \mathrm{Assign}(\mathsf{Y}^{(t)}, \mathbf{R}, z)$
16: $\qquad \qquad \Gamma_{t+1} \leftarrow \Gamma_{t+1} \cup \{ (\mathsf{X}_z^{(t+1)}, \mathsf{Y}_z^{(t+1)}) \}$
17: $\qquad \quad$ **end for**
18: $\qquad \quad \triangleright \mathrm{Assign}(\mathsf{S}, \mathbf{M}, z) = \{ s \in \mathsf{S} \mid \arg\max_{z'} \mathbf{M}_{sz'} = z \}$
19: $\qquad$ **end if**
20: $\quad$ **end for**
21: $\quad t \leftarrow t + 1$
22: **end while**
23: **Output:** $\Gamma_\kappa = \{ (\mathbf{x}_i, T(\mathbf{x}_i))_{i=1}^n \}$ $\qquad \triangleright$ Mapped pairs.

---

factors specific to this $q \in [\rho_t]$:

$$(\mathbf{Q}, \mathbf{R}) \leftarrow \mathrm{LROT}(\mu_{\mathsf{X}_q^{(t)}}, \mu_{\mathsf{Y}_q^{(t)}}, \mathbf{g}_{t+1}). \tag{9}$$

For each $q \in [\rho_t]$ we use the $\mathbf{Q}, \mathbf{R}$ from (9) to co-cluster $\mathsf{X}_q^{(t)}$ with $\mathsf{Y}_q^{(t)}$ using $r_{t+1}$ labels. Within this pair, each $\mathbf{x}_i \in \mathsf{X}_q^{(t)}$ is assigned a label $z \in [r_{t+1}]$ by taking the argmax over the $i$-th row of $\mathbf{Q}$, and likewise each $\mathbf{y}_j \in \mathsf{Y}_q^{(t)}$ is assigned the argmax over the $j$-th row of $\mathbf{R}$. This corresponds to the Assign step in Algorithm 1, and coincides with the hard assignment of $\mathsf{q}^\star$ and $\mathsf{r}^\star$ for an optimal $(\mathbf{Q}^*, \mathbf{R}^*)$ (Lemma B.5).

The uniform constraint $\mathbf{g} = \mathbf{1}_{r_{t+1}}/r_{t+1}$ in (7) enforces an even split of the dataset, which by Lemma B.5 ensures a partition at optimality (for $r_t = 2$). Repeating for all $q \in [\rho_t]$, one obtains a co-clustering with $r_{t+1}$ components within each co-cluster at the previous scale, leading to a total of $\rho_{t+1} = r_{t+1} \rho_t$ co-clusters at scale $t + 1$ (Fig. 1). If the base-case rank $r_{\mathrm{base}}$ is one, Algorithm 1 returns a bijection between $\mathsf{X}$ and $\mathsf{Y}$ as a collection of $n$ tuples.

Note that Hierarchical Refinement defines an implicit hierarchy of block-couplings at each scale $t$.

---

**Definition 3.3** (Hierarchical block-coupling). For each scale $t \in [\kappa]$, given the Hierarchical Refinement co-cluster partition $\Gamma_t$, the *hierarchical block-coupling* at scale $t$ is defined by the matrix

$$\mathbf{P}_{ij}^{(t)} := \frac{\rho_t}{n^2} \sum_{q=1}^{\rho_t} \delta_{(\mathbf{x}_i, \mathbf{y}_j) \in \Gamma_{t,q}}, \tag{10}$$

Without loss of generality, $\mathbf{P}^{(t)}$ may be block diagonalized into $\rho_t$ square blocks, as discussed in Appendix B (see Equation (S13)). By Proposition 3.1, for any rank-schedule $(r_j)_{j=1}^\kappa$ satisfying Monge separability, the final $\mathbf{P}^{(\kappa)}$ corresponds to an optimal coupling supported on the graph of the Monge map $T^\star$, $\mathbf{P}^{(\kappa)} := (\mathrm{id} \times T^\star)_\sharp \mu_{\mathsf{X}}$. While these intermediate couplings are never instantiated, one can still use them to define a transport cost $\langle \mathbf{C}, \mathbf{P}^{(t)} \rangle$ at each scale. In Appendix B.8, we show the following bounds on the cost difference across scales.

**Proposition 3.4.** *Let $c(\cdot, \cdot)$ be a strictly-convex and Lipschitz cost function, let $(r_1, r_2, \cdots, r_\kappa)$ be a rank-schedule, and let $\mathbf{P}^{(t)}$ denote the coupling defined in (10), obtained from step $t$ of Algorithm 1. Define $\Delta_{t,t+1} = \langle \mathbf{C}, \mathbf{P}^{(t)} \rangle_F - \langle \mathbf{C}, \mathbf{P}^{(t+1)} \rangle_F$. Then,*

$$0 \leq \Delta_{t,t+1} \leq \|\nabla c\|_\infty \frac{1}{\rho_t} \sum_{q=1}^{\rho_t} \mathrm{diam}(\Gamma_{t,q}), \tag{11}$$

*where $q$ indexes co-clusters $\Gamma_{t,q}$ at scale $t$, defined in (8).*

Thus, the lower-bound implies that each step of refinement improves the coarse partition, and the upper-bound implies that the difference in solution value is bounded above by a factor depending on the Lipschitz constant and the mean diameter of the coarse partitions at each level $t$. The proof of Proposition 3.4 roughly follows that of Proposition 1 of (Gerber & Maggioni, 2017). In Remark B.9, we discuss how Proposition 3.4 compares, noting that our result makes fewer geometric assumptions on our multiscale partitions $(\mathsf{X}^{(t)})_{t=1}^\kappa$ and $(\mathsf{Y}^{(t)})_{t=1}^\kappa$ and therefore does not quantify the rate of decay of $\mathrm{diam}(\Gamma_{t,q})$.

### 3.3. On the Rank-Annealing Schedule

As observed by (Forrow et al., 2019; Scetbon et al., 2021), rank behaves like a temperature parameter, inverse to the strength $\epsilon$ of entropy regularization. The correspondence between small $\epsilon$ and large rank implies that annealing in the parameter $\epsilon$ is, from the perspective of rank, analogous to initializing the optimization at a low-rank coupling, and then gradually increasing the rank constraint from low to full. In Hierarchical Refinement, this gradual rank increase is accomplished implicitly. At each scale $t = 1, \ldots, \kappa$ the implicit coupling $\mathbf{P}^{(t)}$ is made explicit in the hierarchical block coupling defined in equation (10). A rank-annealing schedule $(r_1, \ldots, r_\kappa)$ describes the sequence of multiplicative factors by which the rank of this explicit coupling will increase

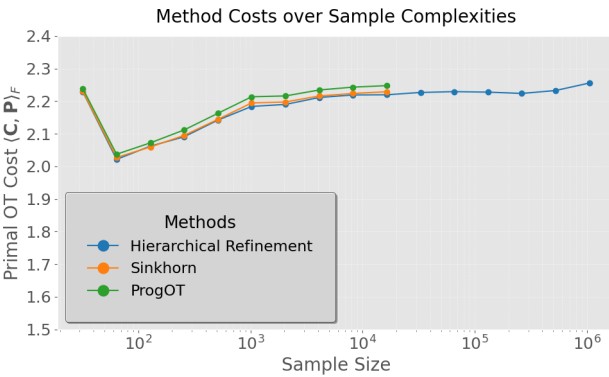

*Figure 2.* Primal OT cost for varying sample size on the synthetic half-moon S-curve dataset of (Buzun et al., 2024) for `HiRef`, Sinkhorn, and ProgOT

at successive scales. The partial products of these, denoted $(\rho_1, \ldots, \rho_\kappa)$, are the ranks of the couplings $\mathbf{P}^{(1)}, \ldots, \mathbf{P}^{(\kappa)}$. Note that small values of $r_i$ generate coarse partitions of the points at the next scale, while large values of $r_i$ generate finer partitions at the next scale.

We now turn to the question of how to efficiently choose such a schedule under given memory constraints. For an integer $n$, Algorithm 1 has log-linear complexity for depth $\kappa = \log_r n$ (Section 3.4). However, the large constants required by low-rank OT in practice encourage minimizing the number of calls to LROT as a subroutine, so that if memory permits, it may be advantageous to decrease the depth by storing couplings of higher rank. If desired, memory constraints can be enforced by imposing a maximum rank $r_{\max} \geq r_t$ for all $t \in [\kappa]$ to ensure Hierarchical Refinement only requires $O(nr_{\max})$ space at each step. Thus, we seek factorizations with *minimal* partial sums of ranks while remaining below a desired memory-capacity:

$$\min_{(r_i)_{i=1}^\kappa} \sum_{j=1}^\kappa \rho_j \quad \text{s.t.} \quad \rho_\kappa = n, \quad r_i \leq r_{\max}. \quad (12)$$

The above optimization assumes a base-rank $r_{\text{base}}$ of 1; we describe how to handle the general case in Appendix E.1. Importantly, the recursive structure $\min_{(r_i)_{i=1}^\kappa} \sum_{j=1}^\kappa \rho_j = \min_{(r_i)_{i=1}^\kappa} \left( r_1 + r_1 \sum_{j=2}^\kappa \prod_{i=2}^j r_i \right)$ enables a dynamic programming approach to (12), storing a table of factors up to $r_{\max}$ to optimize (12) in $O(r_{\max}\kappa n)$ time. Assuming $\kappa, r_{\max}$ are small constants chosen to ensure that all matrices can fit within memory, determining the optimal rank-schedule with respect to $\kappa, n, r_{\max}$ is a simple linear-time procedure.

### 3.4. Complexity and Scaling of Hierarchical Refinement

For two datasets $\mathsf{X}, \mathsf{Y}$ of size $n$, the space complexity of Hierarchical Refinement is $\Theta(n)$, since at each level, one

must store $\Gamma_t$ which is a set of subsets of $\mathsf{X}$ and $\mathsf{Y}$. To derive the time-complexity of Hierarchical Refinement, note that if $n = r^k$, a rank-$r$ schedule at each layer requires $\frac{n}{r}$ instances of LROT over rapidly decaying dataset sizes. The complexity of low-rank OT (Scetbon et al., 2021; 2022; Halmos et al., 2024) is linear $(Kn)$ for a constant $K = O(BLrd)$ dependent on $B$ the number of inner Sinkhorn (Halmos et al., 2024) or Dykstra (Scetbon et al., 2021) iterations, $L$ the number of mirror-descent steps, $r$ the rank of the coupling, and $d$ the rank of the factorization of the cost matrix $\mathbf{C}$. In this setting, for $n$ a power of $r$, the runtime of Algorithm 1 is given by the sum $r^0\Theta(n) + r^1\Theta(\frac{n}{r}) + \ldots + r^{i-1}\Theta(\frac{n}{r^{i-1}}) = \Theta(ndr\log_r n)$ for $i = \log_r n$, achieving *linear* space with *log-linear* time for constant ranks $r, d$.

In cases where the cost matrix does not admit a low-rank factorization $\mathbf{C} = \mathbf{U}\mathbf{V}^\top$, i.e., when $d = O(n)$, one requires $\Theta(n^2)$ space to store the cost matrix and Hierarchical Refinement exhibits time complexity $\tilde{O}(n^2)$, as in Sinkhorn. For kernel costs such as squared Euclidean cost, as noted in (Scetbon et al., 2021), one may efficiently compute a $(d+2)$ dimensional factorization where $d$ is the ambient dimension, to achieve log-linear scaling with exact distances. We also use the sample-linear algorithm of (Indyk et al., 2019) to compute approximate factorizations for distances $c(\cdot, \cdot)$ satisfying metric properties such as the triangle inequality (e.g. Euclidean distance, see Appendix E.1). At each level, pairing such sample-linear approximations with each low-rank step only requires $O(n\log_d n)$ time. We observe this scaling empirically, as reported in Fig. S2.

## 4. Experiments

We benchmark Hierarchical Refinement (`HiRef`) against the full-rank OT methods Sinkhorn (Cuturi, 2013), ProgOT (Kassraie et al., 2024), and mini-batch OT (Genevay et al., 2018; Fatras et al., 2020; 2021b). We additionally benchmark against the low-rank OT methods LOT (Scetbon et al., 2021) and FRLC (Halmos et al., 2024). We use the default implementations of Sinkhorn, ProgOT, and LOT in the high-performance `ott-jax` library (Cuturi et al., 2022). In particular, Sinkhorn is run with the default entropy regularization parameter of $\epsilon = 0.05$. We also benchmark against the multiscale method MOP (Gerber & Maggioni, 2017), which requires multiscale partitions of the input datasets – akin to a family of dyadic cubes across scales – to compute alignments. This leads to a transport cost that depends on the choice of this partition. For simplicity, we choose the default partitions of MOP which are computed from the GMRA (Geometric Multi-Resolution Analysis) R package.

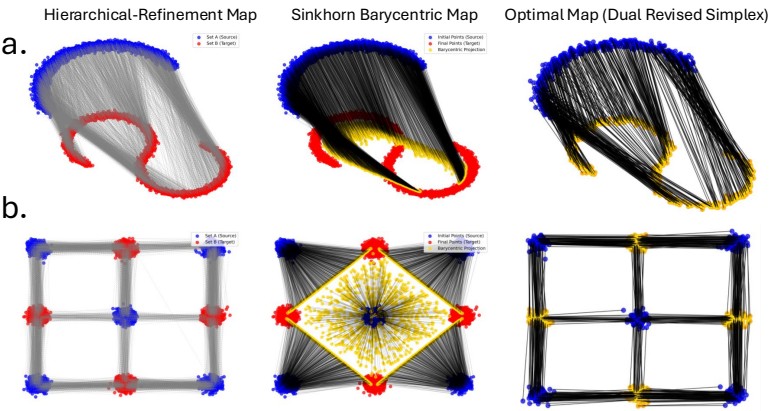

*Figure 3.* Comparison of the Hierarchical Refinement Mapping, the Sinkhorn Barycentric Map, and an optimal map computing using dual revised simplex for the **a.** Half-moon and S-curve dataset (Buzun et al., 2024) of 4096 points (512 points for dual revised simplex) and **b.** Checkerboard dataset (Makkuva et al., 2020).

## 4.1. Evaluation on Synthetic Datasets.

We first evaluate the performance of Hierarchical Refinement against optimal transport methods returning primal couplings, namely Sinkhorn (Cuturi, 2013) (as implemented in ott-jax (Cuturi et al., 2022)) and ProgOT (Kassraie et al., 2024). We evaluate the methods with respect to the Wasserstein-1 and Wasserstein-2 distance on an alignment of 1024 pairs of samples on the Checkerboard (Makkuva et al., 2020), MAFMoons and Rings (Buzun et al., 2024), and Half-Moon and S-Curve (Buzun et al., 2024) synthetic datasets (Fig. 3, Table S6).

All methods are similarly effective at minimizing the primal OT cost $\langle \mathbf{C}, \mathbf{P} \rangle_F$, with small absolute difference in cost between the final couplings. Hierarchical Refinement achieves slightly lower primal cost on 4 out of the 6 evaluations. Notably, there is a massive difference in the number of non-zero entries (defined as entries $\mathbf{P}_{ij} > 10^{-8}$) in the couplings output by HiRef, Sinkhorn, and ProgOT (Table S3). Specifically, across the experiments HiRef outputs a bijection with exactly 1024 non-zero elements in the coupling matrix, equal to the number of aligned samples. In constrast, Sinkhorn and ProgOT output couplings with 624733 to 678720 and 271087 to 337258 non-zero entries.

We evaluate the scalability of Hierarchical Refinement relative to other full-rank solvers on varying numbers of samples from the Half Moon & S-Curve (Buzun et al., 2024) synthetic dataset. We vary the rank from $2^5 = 32$ (64 points aligned) up to $2^{20} = 1048576$ points (2097152 points aligned) in $\mathbb{R}^2$, the latter dataset of a size that is beyond the capabilities of current optimal transport solvers. We observe that Sinkhorn (Cuturi, 2013) and ProgOT – methods which produce dense mappings – require a coupling matrix with $O(n^2)$ non-zero entries and thus run only up

to 16384 points. HiRef yields solutions with comparable primal cost to ProgOT and Sinkhorn on the sample sizes where all methods run.

We also find that HiRef achieves an OT cost that is competitive with the dual revised simplex solver (Huangfu & Hall, 2018), a solver which only scales up to 512 points (Table S4). This solver computes an *optimal* coupling, unlike ProgOT and Sinkhorn which rely on entropic regularization. While we benchmark Sinkhorn in place of mini-batch OT on the synthetic datasets due to their limited complexity, we also evaluate the multi-scale method MOP on the 512 point instance (Table S4). Although MOP outputs a fast approximation to optimal transport, its primal cost on the Checkerboard (Makkuva et al., 2020) dataset is twice as high as that of the other methods, and it performs significantly worse on the MAF Moons & Rings and Half Moon & S-Curve datasets (Buzun et al., 2024).

Lastly, we observe that Hierarchical Refinement scales to over a million points, two orders of magnitude greater than ProgOT and Sinkhorn, two full-rank OT methods that compute global alignments. We find HiRef scales linearly with the size of the problem instance (Fig. S2a) in contrast to the quadratic scaling in time complexity of Sinkhorn (Fig. S2b).

## 4.2. Large-scale Matching Problems and Transcriptomics

Recently, optimal transport has been applied to single-cell and spatial transcriptomics datasets to compute couplings between cells taken from different timepoints from developmental processes or perturbations (Schiebinger et al., 2019; Lavenant et al., 2024; Bunne et al., 2022; Huizing et al., 2024; Halmos et al., 2025b; Klein et al., 2025). However, the size of current datasets (Chen et al., 2022) (>100k cells)

*Table 1.* Cost Values $\langle \mathbf{C}, \mathbf{P} \rangle_F$ Across Later Embryonic Stages

| Method | E12-13.5 | E13-14.5 | E14-15.5 | E15-16.5 |
|--------|----------|----------|----------|----------|
| HiRef | **14.35** | **13.78** | **14.29** | **12.79** |
| Sinkhorn | - | - | - | - |
| MB 128 | 14.86 | 14.14 | 14.75 | 13.32 |
| MB 1024 | 14.45 | 13.86 | 14.43 | 12.91 |
| FRLC | 15.47 | 14.64 | 15.51 | 14.00 |

has exceeded the capacity of existing full-rank solvers, requiring low-rank approximations of the coupling (Scetbon et al., 2023; Klein et al., 2025; Halmos et al., 2025a) to produce alignments.

We evaluate whether the full-rank solver of Hierarchical Refinement exhibits competitive alignments for such datasets. Specifically, we analyze the mouse organogenesis spatiotemporal transcriptomic atlas (MOSTA) datasets, which include spatial transcriptomics data from mouse embryos at successive 1-day time-intervals with increasing number $n$ of cells at each stage: E9.5 ($n = 5913$), E10.5 ($n = 18408$), E11.5 ($n = 30124$), E12.5 ($n = 51365$), E13.5 ($n = 77369$), E14.5 ($n = 102519$), E15.5 ($n = 113350$), and E16.5 ($n = 121767$). For the cost we use the Euclidean distance $\mathbf{C}_{ij} = \|\mathbf{x}_i - \mathbf{y}_j\|_2$ in 60-dimensional PCA space of expression vectors, so $\mathbf{x}_i, \mathbf{y}_j \in \mathbb{R}^{60}$.

Sinkhorn and ProgOT are unable to produce alignments for the stages beyond E10.5 ($n = 18408$ cells), whereas HiRef, the low-rank solvers, and mini-batch OT (batch-sizes $B = 128$ to $B = 2048$) are able to continue scaling to $> 10^5$ (Table 1, Table S6). We observe that the Kantorovich cost of HiRef is consistently lower than all other methods for all timepoints (Table 1, Table S6).

HiRef achieves a substantially lower cost than the low-rank solvers FRLC and LOT for rank $r = 40$, even though HiRef relies on low-rank optimal transport (FRLC) as a subroutine. This result underscores the empirical trend observed in Fig. S3, where the refinement step of HiRef progressively decreases the primal cost of coarser low-rank couplings (Proposition 3.4). While the mini-batch solvers exhibit competitive scaling up to the last pair, the primal cost of mini-batch is higher for all tested batch-sizes (Table S6). Unlike HiRef, mini-batch OT does not compute a global alignment and exhibits batch-size dependent error.

### 4.3. MERFISH Brain Atlas Alignment

We ran HiRef on two slices of MERFISH Mouse Brain Receptor Map data from Vizgen to test whether HiRef can produce biologically valid alignments using the *only* spatial densities of each tissue. These spatial transcriptomics data consist of spatial and gene expression measurements at individual spots in three full coronal slices across three bio-

logical replicates. Our "source" dataset $(\mathbf{X}^1, \mathbf{S}^1)$ is replicate 3 of slice 2, while our "target" dataset $(\mathbf{X}^2, \mathbf{S}^2)$ is replicate 2 of slice 2, following the expression transfer task described (Clifton et al., 2023) between these two slices. Each dataset has roughly 84k spots, where memory constraints prohibit instantiation a full-rank alignment as a matrix. Thus, solvers such as Sinkhorn (Cuturi, 2013) and ProgOT (Kassraie et al., 2024) are unable to run on the dataset.

We use only spatial information when building a map between the two slices, using the spatial Euclidean cost $\mathbf{C}_{ij} := \|\mathbf{s}_i^1 - \mathbf{s}_j^2\|_2$, after registering spatial coordinates $\mathbf{S}^1 = \{\mathbf{s}_i^1\}_{i=1}^n$ and $\mathbf{S}^2 = \{\mathbf{s}_i^2\}_{i=1}^n$ with an affine transformation. We gauged the quality of the HiRef alignment (Fig. 4a), using gene expression abundances of five "spatially-varying" genes. Specifically, we observe that expression vector $\mathbf{v}^1$ of gene *Slc17a7* in the source slice (Fig. 4b) when transferred to target slice through the bijective mapping output by HiRef, denoted as $\hat{\mathbf{v}}$ (Fig. 4c), closely matches the observed expression vector $\mathbf{v}^2$ of *Slc17a7* in the target slice (Fig. 4d) with cosine similarity equal to 0.8098. For genes *Slc17a7*, *Grm4*, *Olig1*, *Gad1*, *Peg10*, the corresponding cosine similarities between the transferred and observed expression vectors are 0.8098, 0.7959, 0.7526, 0.4932, 0.6015, respectively.

For comparison, we also ran the low-rank methods FRLC (Halmos et al., 2024) and LOT (Scetbon et al., 2021) with and without subsampling, reporting their best scores, as discussed in Section D.3. For the gene *Slc17a7*, FRLC's cosine similarity was 0.2373, while LOT's cosine similarity was 0.3390. For all five genes *Slc17a7*, *Grm4*, *Olig1*, *Gad1*, *Peg10*, FRLC's scores were (0.2373, 0.2124, 0.1929, 0.0963, 0.1550, respectively, while LOT's scores were 0.3390, 0.2712, 0.3186, 0.1666, 0.1080. Across all five genes HiRef's scores were at least twice those of FRLC or LOT (Table S7) with gene abundances shown in Fig. S1. On the same task, we compared against MOP, the method of (Gerber & Maggioni, 2017), whose scores for the five genes were: (0.5211, 0.4714, 0.5972, 0.3571, 0.2719). Finally, we also benchmarked against mini-batch OT using batch sizes ranging from 128 to 2048 in powers of two, whose best scores (0.7434, 0.7822, 0.7056, 0.4912, 0.5683) were more comparable to that of the performance of HiRef. Across all methods and genes compared in Table S7, HiRef had greatest cosine similarity scores in the expression transfer task, while also having lowest transport cost. Further experimental details are in Section D.3.

### 4.4. ImageNet Alignment

We demonstrate the scalability of Hierarchical Refinement on a large-scale and high-dimensional dataset by aligning 2048-dimensional embeddings of 1.281 million images from the ImageNet ILSVRC dataset (Deng et al., 2009;

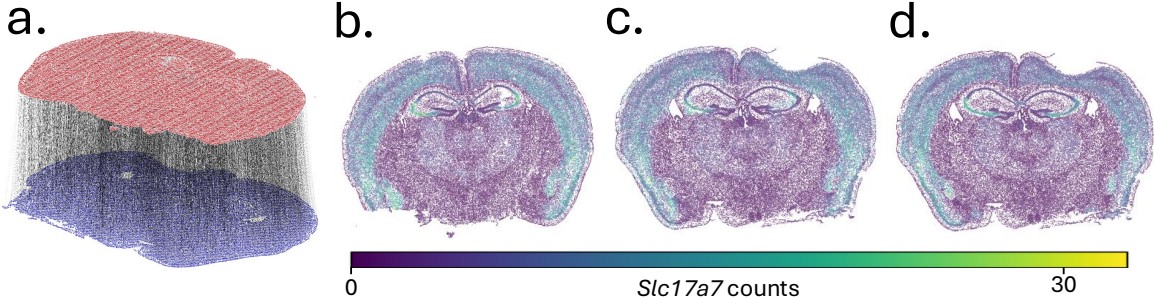

**a.** **b.** **c.** **d.**

0      *Slc17a7* counts      30

*Figure 4.* **a.** Hierarchical Refinement alignment on MERFISH mouse brain data, using only spatial coordinates. **b.** Abundance $\mathbf{v}^1$ of gene *Slc17a7* in the source slice. **c.** Predicted *Slc17a7* abundance $\hat{\mathbf{v}}$ from the source slice to the target slice, through the `HiRef` coupling. **d.** Abundance $\mathbf{v}^2$ of the same gene in the target slice. Transferred abundances $\hat{\mathbf{v}}$ have cosine similarity 0.8098 with true abundances $\mathbf{v}^2$ in the target.

*Table 2.* Cost Values $\langle \mathbf{C}, \mathbf{P} \rangle_F$ for ImageNet (Deng et al., 2009; Russakovsky et al., 2015) Alignment Task.

| Method | HiRef | MB 128 | MB 256 | MB 512 | MB 1024 | FRLC |
|--------|-------|--------|--------|--------|---------|------|
| **OT Cost** | **18.97** | 21.89 | 21.11 | 20.34 | 19.58 | 24.12 |

Russakovsky et al., 2015). Each image is embedded using the ResNet50 architecture (He et al., 2016), and we construct two datasets, X and Y, by taking a random 50:50 split of the embedded images. We align X and Y using `HiRef`, FRLC, and mini-batch OT with batch-sizes ranging from $B = 128$ to $B = 1024$. ProgOT, Sinkhorn, and LOT could not be run on the datasets due to memory constraints. `HiRef` yielded a primal OT cost of 18.974, while FRLC (Halmos et al., 2024) solution had a primal OT cost of 24.119 for rank $r = 40$ and mini-batch OT has costs of 21.89 ($B = 128$) to 19.58 ($B = 1024$) (Table 2).

## 5. Discussion

Hierarchical Refinement computes the Monge map between large-scale datasets in linear space, but has several limitations. First, we currently assume that the datasets X and Y have the same number of samples. In many machine learning applications, this is not a limiting factor, as one generally seeks to pair an equal number of source points $\mathbf{x}$ to target points $\mathbf{y}$. Second, while Hierarchical Refinement scales linearly in space and log-linearly in time, it still involves a constant dependent on the low-rank OT sub-procedure used – this underscores the need to accelerate and stabilize low-rank OT solvers further (Scetbon & Cuturi, 2022; Halmos et al., 2024). Finally, while Hierarchical Refinement guarantees an optimal solution given an optimal black-box low-rank solver (Proposition 3.1), the low-rank solvers (Scetbon et al., 2022; Halmos et al., 2024) used in practice are not necessarily optimal, owing to the non-convexity of low-rank problems.

Optimal transport has been successfully applied in deep learning frameworks, such as OT flow-matching (Tong et al., 2024), computer vision and point cloud registration, (Yu et al., 2021; Qin et al., 2022), among many others. The mini-batch procedure used to train many of these methods involves sampling two datasets $\mathsf{X}_B \sim \mu$ and $\mathsf{Y}_B \sim \nu$ with batch-size $B$ and aligning them with Sinkhorn at every training iteration. `HiRef` suggests an alternative approach: one can precompute millions of *globally aligned* pairs and then sample $\mathsf{X}_B \sim \mu$ and the optimal mapping $T(\mathsf{X}_B) \sim \nu$ by indexing into these precomputed pairs. This approach applies to any loss function dependent on an OT alignment.

Hierarchical Refinement may also be useful in neural OT approaches which learn a continuous Monge map between the densities of two datasets. For example, (Seguy et al., 2018) minimize a loss $\min_\theta \frac{1}{2}\mathbb{E}_\mu \|T_\theta(\mathbf{x}_i) - T(\mathbf{x}_i)\|_2^2$ between a neural network $T_\theta$ with parameters $\theta$ and a Monge map $T$ over samples $\mathbf{x}_i \sim \mu$ (Remark B.11). Thus, the procedure outlined above may be used to directly regress a neural network $T_\theta$ on the Monge map $T$ without the bias of mini-batching or entropy.

## 6. Conclusion

We introduce Hierarchical Refinement (`HiRef`), an algorithm to solve optimal transport with linear complexity in the number of points, making sparse, full-rank optimal transport feasible for large-scale datasets. Our algorithm leverages that low-rank optimal transport co-clusters points with their image under the Monge map, refining bijections between partitions of each dataset across a hierarchy of scales, down to a bijective Monge map between the datasets at the finest scale. Hierarchical Refinement couplings achieve comparable primal cost to couplings obtained through full-rank entropic solvers, and scales to datasets with over a million points, opening the door to applications previously infeasible for optimal transport.

## Acknowledgements

We thank Henri Schmidt for many helpful conversations. This research was supported by NIH/NCI grant U24CA248453 to B.J.R. J.G. is supported by the Schmidt DataX Fund at Princeton University made possible through a major gift from the Schmidt Futures Foundation.

## Impact Statement

Optimal transport has emerged as a powerful tool in generative modeling, yet its widespread use has been limited by scalability constraints. `HiRef` overcomes this limitation by enabling the application of OT to datasets with millions of points. This advancement paves the way for integrating OT into large-scale deep generative models and modern vision and language tasks.

As with any computational tool which may advance large-scale generative modeling, there are potential issues with bias in training datasets and a possibility of misuse. Use of `HiRef` in applications should be careful and transparent about these risks and utilize appropriate mitigation strategies.

## Code Availability

Our implementation of Hierarchical Refinement is available at https://github.com/raphael-group/HiRef.

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

# A. Hierarchical-Refinement Algorithm

---

**Algorithm 2** Hierarchical Refinement for Full-Rank OT

---

**Require: Datasets** $\mathsf{X} = \{\mathbf{x}_i\}_{i=1}^n$, $\quad$ $\mathsf{Y} = \{\mathbf{y}_i\}_{i=1}^n$; **Low-rank OT solver** $\mathrm{LROT}(\cdot)$; **Rank schedule** $(r_1, r_2, \ldots, r_\kappa)$;
$\qquad$ **Base rank** $r_{\text{base}} = \frac{n}{\prod_{t=1}^\kappa r_t}$ (e.g. 1).
$\qquad$ **Initialize:**
1: $t \leftarrow 0, \Gamma_0 \leftarrow \{(\mathsf{X}, \mathsf{Y})\}$
2: **while** $\exists (\mathsf{X}^{(t)}, \mathsf{Y}^{(t)}) \in \Gamma_t$ **such that**
3: $\qquad\qquad \min\{|\mathsf{X}^{(t)}|, |\mathsf{Y}^{(t)}|\} > r_{\text{base}}$ **do**
4: $\quad$ $\Gamma_{t+1} \leftarrow \varnothing$
5: $\quad$ **for** $(\mathsf{X}_q^{(t)}, \mathsf{Y}_q^{(t)}) \in \Gamma_t$ **do**
6: $\qquad$ **if** $\min\{|\mathsf{X}_q^{(t)}|, |\mathsf{Y}_q^{(t)}|\} \leq r_{\text{base}}$ **then**
7: $\qquad\quad$ $\Gamma_{t+1} \leftarrow \Gamma_{t+1} \cup \{(\mathsf{X}_q^{(t)}, \mathsf{Y}_q^{(t)})\}$
8: $\qquad$ **else**
9: $\qquad\quad$ $\mu_{\mathsf{X}_q^{(t)}} = \frac{1}{|\mathsf{X}_q^{(t)}|} \sum_{\mathbf{x} \in \mathsf{X}_q^{(t)}} \delta_{\mathbf{x}}$
10: $\qquad\quad$ $\mu_{\mathsf{Y}_q^{(t)}} = \frac{1}{|\mathsf{Y}_q^{(t)}|} \sum_{\mathbf{y} \in \mathsf{Y}_q^{(t)}} \delta_{\mathbf{y}}.$
11: $\qquad\quad$ $\mathbf{g}_{t+1} \leftarrow (1/r_{t+1})\mathbf{1}_{r_{t+1}}$
12: $\qquad\quad$ $(\mathbf{Q}, \mathbf{R}) \leftarrow \mathrm{LROT}(\mu_{\mathsf{X}_q^{(t)}}, \mu_{\mathsf{Y}_q^{(t)}}, \mathbf{g}_{t+1})$
13: $\qquad\quad$ **for** $z = 1 \rightarrow r_{t+1}$ **do**
14: $\qquad\qquad$ $\mathsf{X}_z^{(t+1)} \leftarrow \mathrm{Assign}(\mathsf{X}^{(t)}, \mathbf{Q}, z)$
15: $\qquad\qquad$ $\mathsf{Y}_z^{(t+1)} \leftarrow \mathrm{Assign}(\mathsf{Y}^{(t)}, \mathbf{R}, z)$
16: $\qquad\qquad$ $\Gamma_{t+1} \leftarrow \Gamma_{t+1} \cup \{(\mathsf{X}_z^{(t+1)}, \mathsf{Y}_z^{(t+1)})\}$
17: $\qquad\quad$ **end for**
18: $\qquad\qquad\qquad\qquad\qquad\qquad\qquad\qquad\qquad\qquad\qquad$ $\triangleright$ $\mathrm{Assign}(\mathsf{S}, \mathbf{M}, z) = \{s \in \mathsf{S} \mid \arg\max_{z'} \mathbf{M}_{sz'} = z\}$
19: $\qquad$ **end if**
20: $\quad$ **end for**
21: $\quad$ $t \leftarrow t + 1$
22: **end while**
23: **Output:** $\Gamma_\kappa = \{(\mathbf{x}_i, T(\mathbf{x}_i))\}$ $\qquad\qquad\qquad\qquad\qquad\qquad\qquad\qquad$ $\triangleright$ Set of refined pairs.

---

# B. Proofs

Datasets $\mathsf{X}$ and $\mathsf{Y}$ are represented as discretely supported probability measures $\mu = \sum_{i=1}^n \mathbf{a}_i \delta_{\mathbf{x}_i}$ and $\nu = \sum_{j=1}^n \mathbf{b}_j \delta_{\mathbf{y}_j}$ for probability vectors $\mathbf{a}, \mathbf{b} \in \Delta_n$, which we assume to be uniform: $\mathbf{a} = \mathbf{b} = \boldsymbol{u}_n = (1/n)\mathbf{1}_n \in \Delta_n$. We form the cost matrix $\mathbf{C}$ defined by

$$\mathbf{C}_{ij} := c(\mathbf{x}_i, \mathbf{y}_j). \tag{S1}$$

In all cases below, we are concerned with the assignment problem (1) for this cost matrix.

Let $\mathbf{perm}(n) = \{\tilde{\mathbf{P}} \in \mathbb{R}^{n \times n} : \tilde{\mathbf{P}}\mathbf{1}_n = \tilde{\mathbf{P}}^\top \mathbf{1}_n = (1/n)\mathbf{1}_n\}$ denote the set of (scaled) $n \times n$ permutation matrices. By the Birkhoff-von Neumann theorem (Birkhoff, 1946), an optimal solution to the $n \times n$ assignment problem is attained at a permutation matrix in $\mathbf{perm}(n)$.

**Definition B.1.** Say that cost matrix $\mathbf{C} \in \mathbb{R}^{n \times n}$ is *Monge rotated* if the identity matrix $\mathbf{I}$ is a solution to the assignment problem associated to $\mathbf{C}$, i.e.

$$\mathbf{I} \in \arg\min_{\mathbf{P} \in \mathbf{perm}(n)} \langle \mathbf{C}, \mathbf{P} \rangle.$$

For arbitrary cost matrix $\mathbf{C} \in \mathbb{R}^{n \times n}$, let $\mathbf{P}^\dagger \in \arg\min_{\mathbf{P} \in \mathbf{perm}(n)} \langle \mathbf{C}, \mathbf{P} \rangle_F$, and note that the column-permuted cost matrix $\mathbf{C}^\dagger := \mathbf{C}\mathbf{P}^{\dagger,\top}$ is Monge rotated by construction. This is a consequence of the following identity, which holds for any

permutation $\tilde{\mathbf{P}} \in \mathbf{perm}(n)$.

$$\begin{aligned}
\langle \mathbf{C}, \mathbf{P} \rangle_F &= \mathbf{tr}(\mathbf{C}^\top \mathbf{P}) \\
&= \mathbf{tr}(\tilde{\mathbf{P}}^{-1} \tilde{\mathbf{P}} \mathbf{C}^\top \mathbf{P}) \\
&= \mathbf{tr}(\tilde{\mathbf{P}} \mathbf{C}^\top \mathbf{P} \tilde{\mathbf{P}}^\top) = \langle \mathbf{C} \tilde{\mathbf{P}}^\top, \mathbf{P} \tilde{\mathbf{P}}^\top \rangle_F.
\end{aligned} \tag{S2}$$

Let $\Pi(\boldsymbol{u}_n, \boldsymbol{u}_r) \equiv \Pi_{\boldsymbol{u}_n, \boldsymbol{u}_2}$ denote the transport polytope between two uniform measures. For $\mathbf{Q} \in \Pi(\boldsymbol{u}_n, \boldsymbol{u}_r)$, say that a row of $\mathbf{Q}$ is *soft* if at least two of its entries are positive, and call the row *hard* otherwise. For rank $r \ll n$ such that $r$ divides $n$, let $\Pi_\bullet(\boldsymbol{u}_n, \boldsymbol{u}_r)$ be the subset of $\Pi(\boldsymbol{u}_n, \boldsymbol{u}_r)$ consisting of transport plans $\mathbf{Q}$ with only hard rows. Below, we consider two low-rank OT problems associated to $\mathbf{C}^\dagger$. The first low-rank problem considered is

$$\min_{\mathbf{Q}, \mathbf{R} \in \Pi_\bullet(\boldsymbol{u}_n, \boldsymbol{u}_r)} \langle \mathbf{C}^\dagger, \mathbf{Q} \mathbf{R}^\top \rangle_F. \tag{S3}$$

while the second low-rank problem considered is restricted to symmetric couplings:

$$\min_{\mathbf{Q} \in \Pi_\bullet(\boldsymbol{u}_n, \boldsymbol{u}_r)} \langle \mathbf{C}^\dagger, \mathbf{Q} \mathbf{Q}^\top \rangle_F. \tag{S4}$$

In either case, we have omitted the constant factor of $r$ coming from $\mathrm{diag}(1/\boldsymbol{u}_r)$. We next introduce a technical condition on $\mathbf{C}$. Let $\mathbf{C} \in \mathbb{R}^{n \times n}$ be a cost matrix and let $\mathbf{P}^\dagger \in \arg\min_{\mathbf{P} \in \mathbf{perm}(n)} \langle \mathbf{C}, \mathbf{P} \rangle$ corresponding to permutation $\sigma^\dagger : [n] \to [n], \sigma^\dagger \in \mathfrak{S}_n$. Given partitions $\mathcal{I} = \{I_1, \dots, I_r\}$ and $\mathcal{J} = \{J_1, \dots, J_r\}$ of $[n]$ and $a, b \in [r]$, define the cost between two sets $I_a, J_b$ to be

$$\mathbf{C}_{I_a, J_b} := \sum_{i \in I_a, j \in J_b} \mathbf{C}_{i \sigma^\dagger(j)}. \tag{S5}$$

We call partition $\mathcal{I}$ *balanced* if each block $I_a$ of $\mathcal{I}$ has the same number of elements, $|I_a| = (n/r)$.

**Definition B.2.** Cost matrix $\mathbf{C} \in \mathbb{R}^{n \times n}$ is *$r$-Monge separable* if there exists a balanced partition $\mathcal{I}^\star = \{I_k^\star\}_{k=1}^r$, such that for any two permutations $\pi_1, \pi_2 \in \mathfrak{S}_n$, one has

$$\sum_{k=1}^r \mathbf{C}_{I_k^\star, I_k^\star} \leq \sum_{k=1}^r \mathbf{C}_{\pi_1(I_k^\star), \pi_2(I_k^\star)}. \tag{S6}$$

We say that $\mathbf{C}$ is *strictly $r$-Monge separable* if (S6) holds with strict inequality $(<)$ for any $\pi_1(I_k^\star) \neq \pi_2(I_k^\star)$.

One interesting feature of this definition is that while the sum is over $r \leq n$ terms, where it may occur that $r \ll n$, this inequality must hold over all permutations $\pi_1$ and $\pi_2$ acting on the individual data points, rather than partition blocks. This captures the notion of finding low-rank or low-resolution solutions which are nevertheless compatible with the optimal bijective Monge map.

*Remark* B.3. If $\mathbf{C}$ is $r$-Monge separable, the distinguished partition $\mathcal{I}^\star$ may be represented as $\mathbf{Q}^\star \in \Pi_\bullet(\boldsymbol{u}_n, \boldsymbol{u}_r)$ such that $\mathbf{Q}^\star$ is optimal for (S4) and the pair $(\mathbf{Q}^\star, \mathbf{Q}^\star)$ is optimal for (S3). After proving the next lemma, we will relate $r$-Monge separability to cyclic monotonicity.

**Proposition B.4.** *Let $\mathbf{C} \in \mathbb{R}^{n \times n}$ be strictly $r$-Monge separable. If $\mathbf{Q}^\star, \mathbf{R}^\star \in \arg\min_{\mathbf{Q}, \mathbf{R} \in \Pi_\bullet(\boldsymbol{u}_n, \boldsymbol{u}_2)} \langle \mathbf{C}, \mathbf{Q} \mathbf{R}^\top \rangle$ then, for all $i \in [n]$,*

$$\arg\max_{z \in [r]} \mathbf{Q}_{iz}^\star = \arg\max_{z \in [r]} \mathbf{R}_{\sigma^\dagger(i)z}^\star, \tag{S7}$$

*where $\sigma^\dagger : [n] \to [n]$ is the permutation corresponding to $\mathbf{P}^\dagger \in \arg\min_{\mathbf{P} \in \mathbf{perm}(n)} \langle \mathbf{C}, \mathbf{P} \rangle_F$.*

*Proof.* Let $\sigma^\dagger, \mathbf{P}^\dagger$ be as in the statement of the lemma, and define $\mathbf{C}^\dagger := \mathbf{C} \mathbf{P}^{\dagger, \top}$. The same reasoning as in (S2) implies that if $(\mathbf{Q}^\star, \mathbf{R}^\star) \in \arg\min_{\mathbf{Q}, \mathbf{R} \in \Pi_\bullet(\boldsymbol{u}_n, \boldsymbol{u}_2)} \langle \mathbf{C}, \mathbf{Q} \mathbf{R}^\top \rangle_F$, then

$$(\mathbf{Q}^\star, \mathbf{P}^\dagger \mathbf{R}^\star) \in \arg\min_{\mathbf{Q}, \mathbf{R} \in \Pi_\bullet(\boldsymbol{u}_n, \boldsymbol{u}_2)} \langle \mathbf{C}^\dagger, \mathbf{Q} \mathbf{R}^\top \rangle_F. \tag{S8}$$

The membership (S8) follows from the identities

$$\langle \mathbf{C}^\dagger, \mathbf{Q}^\star \mathbf{R}^\star \mathbf{P}^{\dagger,\top} \rangle_F = \langle \mathbf{C}\mathbf{P}^{\dagger,\top}, \mathbf{Q}^\star \mathbf{R}^{\star,\top} \mathbf{P}^{\dagger,\top} \rangle_F,$$
$$= \mathbf{tr}(\mathbf{P}^\dagger \mathbf{C}^\top \mathbf{Q}^\star \mathbf{R}^{\star,\top} \mathbf{P}^{\dagger,\top}),$$
$$= \mathbf{tr}\, \mathbf{C}^\top \mathbf{Q}^\star \mathbf{R}^{\star,\top} = \langle \mathbf{C}, \mathbf{Q}^\star \mathbf{R}^{\star,\top} \rangle_F.$$

Remark B.3 above follows from the requirement that the variables $\mathbf{Q}, \mathbf{R}$ have all hard rows, and are subject to uniform marginal constraints, so that all non-zero entries of $\mathbf{Q}\mathbf{R}^\top$ have the same value. Thus, if $\mathbf{C}$ is $r$-Monge separable, there exists $\tilde{\mathbf{Q}} \in \Pi_\bullet(\boldsymbol{u}_n, \boldsymbol{u}_2)$ corresponding to distinguished balanced partition $\tilde{\mathcal{I}}$ from Definition B.2 such that

$$(\tilde{\mathbf{Q}}, \tilde{\mathbf{Q}}) \in \operatorname*{arg\,min}_{\mathbf{Q}, \mathbf{R} \in \Pi_\bullet(\boldsymbol{u}_n, \boldsymbol{u}_2)} \langle \mathbf{C}^\dagger, \mathbf{Q}\mathbf{R}^\top \rangle. \tag{S9}$$

Moreover, this pair $(\tilde{\mathbf{Q}}, \tilde{\mathbf{Q}})$ is the unique optimum when $\mathbf{C}$ is strictly $r$-Monge separable. From (S8), (S9), we must have

$$\tilde{\mathbf{Q}} = \mathbf{Q}^\star, \quad \tilde{\mathbf{Q}} = \mathbf{P}^\dagger \mathbf{R},$$

from which (S7) follows immediately. $\qquad\square$

Let us now discuss how the notion of $r$-Monge separability is related to $c$-cyclic monotonicity. Recall that for a cost matrix $\mathbf{C} \in \mathbb{R}^{n \times n}$ derived from ground cost $c$ the support of an optimal plan is $c$-cyclically monotone if for all permutations $\pi : [n] \to [n], \pi \in \mathfrak{S}_n$, one has

$$\sum_{i=1}^n \mathbf{C}_{ii} \le \sum_{i=1}^n \mathbf{C}_{i\pi(i)}. \tag{S10}$$

As it amounts to a reindexing of the sum on the right side of (S10) , one can equivalently define the support of the optimal plan to be $c$-cyclically monotone if for any *pair* of permutations $\pi_1, \pi_2 \in \mathfrak{S}_n$,

$$\sum_{i=1}^n \mathbf{C}_{ii} \le \sum_{i=1}^n \mathbf{C}_{\pi_1(i)\pi_2(i)},$$

from which we see that $c$-cyclical monotonicity is equivalent to $r$-Monge separability with $r = n$.

We next show that the optimal factors $\mathbf{Q}^\star, \mathbf{R}^\star$ for the rank-2 Wasserstein problem given in (5) correspond to hard-partitions of each dataset, so that for this problem the optimal $\mathbf{Q}^\star, \mathbf{R}^\star \in \Pi(\boldsymbol{u}_n, \boldsymbol{u}_2)$ satisfy $\mathbf{Q}^\star, \mathbf{R}^\star \in \Pi_\bullet(\boldsymbol{u}_n, \boldsymbol{u}_r)$. Below, let $\operatorname{supp}_i(\mathbf{Q}^\star) \subset [n]$ be the indices on which column $i$ of $\mathbf{Q}^\star$ is supported, where $i = 1, 2$.

**Lemma B.5.** *Let* $(\mathbf{Q}^\star, \mathbf{R}^\star)$ *be optimal for the rank-2 Wasserstein problem* (5) *subject to the additional constraint that* $\mathbf{a} = \mathbf{b} = \boldsymbol{u}_n$, *and* $\mathbf{g} = \boldsymbol{u}_2$ *are uniform and* $n = m$ *is even. Then,* $(\operatorname{supp}_1(\mathbf{Q}^\star), \operatorname{supp}_2(\mathbf{Q}^\star))$ *is a partition of* $[n]$, *and symmetrically, so is* $(\operatorname{supp}_1(\mathbf{R}^\star), \operatorname{supp}_2(\mathbf{R}^\star))$, *so* $(\mathbf{Q}^\star, \mathbf{R}^\star) \in \Pi_{\bullet(\boldsymbol{u}_n, \boldsymbol{u}_2)}$.

*Proof.* The cost is linear in $(\mathbf{Q}, \mathbf{R})$ respectively: the minimization in each variable given the other fixed can be expressed as

$$\operatorname*{arg\,min}_{\mathbf{Q} \in \Pi(\boldsymbol{u}_n, \boldsymbol{u}_2)} 2 \langle \mathbf{Q}, \mathbf{C}\mathbf{R} \rangle_F, \qquad \operatorname*{arg\,min}_{\mathbf{R} \in \Pi(\boldsymbol{u}_n, \boldsymbol{u}_2)} 2 \langle \mathbf{R}, \mathbf{C}^\top \mathbf{Q} \rangle_F. \tag{S11}$$

Thus for any optimal $\mathbf{Q}^\star$ or $\mathbf{R}^\star$ fixed the minimization in the other variable is a linear optimal transport problem, where by Corollary 2.11 in (De Loera & Kim, 2013) it holds that since the constraint matrix is totally unimodular with marginals integral (on rescaling), the optima $\mathbf{R}^\star$ and $\mathbf{Q}^\star$ must be vertices on the transport polytope $\Pi_{\boldsymbol{u}_n, \boldsymbol{u}_2}$ with integral entries (on rescaling, by $2n$ or $2m$). There are $\le n + 1$ positive entries in any optimal rank $r = 2$ solution (De Loera & Kim, 2013; Peyré & Cuturi, 2019), so that $n$ (resp. $m$) being even and the rescaled rows and columns summing to 2 and $n$ implies that there are exactly $n$ positive entries and thus that the vertices define partitions of $[n]$ and $[m]$. Thus, solutions to S11 satisfy $(\mathbf{Q}^\star, \mathbf{R}^\star) \in \Pi_{\bullet(\boldsymbol{u}_n, \boldsymbol{u}_2)}$. $\qquad\square$

Notably, in the case of an odd number of points $n$ or $m$ this likewise implies that one has a single row which has 2 entries $\begin{pmatrix} 1/2n & 1/2n \end{pmatrix}$, with all other rows of the form $\begin{pmatrix} 0 & 1/n \end{pmatrix}$ or $\begin{pmatrix} 1/n & 0 \end{pmatrix}$ defining a partition of the remaining even subset of size $(n-1)$ or $(m-1)$. In the general case of ranks $r \neq 2$ there are maximally $n + r + 1$ (Peyré & Cuturi, 2019) non-zero edges (so that the graph is acyclic), and for $n \gg r$ the optimal solution remains close to a partition given mild assumptions on $\mathbf{C}$.

Lemma B.5 states optimal low-rank couplings $(\mathbf{Q}^\star, \mathbf{R}^\star)$ for Problem 7 over $\Pi_{(\boldsymbol{u}_n, \boldsymbol{u}_2)}$ are in $\Pi_{\bullet(\boldsymbol{u}_n, \boldsymbol{u}_2)}$. Thus, by Proposition B.4 these solutions co-cluster points $\mathbf{x} \in \mathsf{X}$ with their image under Monge map $T^\star(\mathbf{x})$, supposing the cost is strictly 2-Monge separable (Definition B.2). This co-clustering is in the sense of the clustering functions $\mathsf{q}^\star, \mathsf{r}^\star$ from Proposition 3.1 corresponding to each factor $\mathbf{Q}^\star, \mathbf{R}^\star$. We note that when $\mu$ and $\nu$ are discretely supported measures with supports of equal cardinality, a Monge map, $T^\star : \mathsf{X} \to \mathsf{Y}$, is guaranteed to exist by Theorem 2.7 of (Thorpe, 2018).

**On the Rank Schedule.** At each intermediate scale $t \in [\kappa]$, the *rank-schedule* $(r_1, \dots, r_\kappa)$ determines the effective rank of the coupling computed so far. For each $t \in [\kappa]$, define the *effective rank* at scale $t$ as $\rho_t := \prod_{s=1}^t r_s$. This effective rank corresponds to the number of partitions, which are placed in bijective correspondence

$$\mathsf{X}_q^{(t)} \leftrightarrow \mathsf{Y}_q^{(t)} \quad t \in [\rho_t]. \tag{S12}$$

at the $t$-th step of $\mathtt{HiRef}$. The size of the partitions at scale $t$ is given by $n/\rho_t = |\mathsf{X}^{(t)}| = |\mathsf{Y}^{(t)}|$. Given these preliminaries, we show that for an appropriate rank-schedule Hierarchical Refinement yields optimal transport maps.

**Proposition B.6** (Optimality of Hierarchical Refinement)**.** *Suppose the Monge-map exists between two datasets $\mathsf{X}$, $\mathsf{Y}$ of size $n$. Then there exists a rank-schedule $(r_1, \cdots, r_\kappa)$ which factorizes $n$ such that all size $n/\rho_t$ partitions generated by Hierarchical Refinement at level $t$ satisfy strict $r_{t+1}$-Monge separability (Definition B.2) for $t \in [0 : \kappa - 1]$. For any such rank-schedule, given an optimal black-box low-rank solver over $\Pi_\bullet(\cdot, \cdot)$, Hierarchical Refinement returns the Monge-map.*

*Proof.* For existence, observe that taking $r_1 = n$ implies the statement $\sum_{k=1}^n \mathbf{C}_{I_k^\star, I_k^\star} \leq \sum_{k=1}^n \mathbf{C}_{\pi_1(I_k^\star), \pi_2(I_k^\star)}$. For partitions $I_k$ of size one, this is equivalent to the statement of $c$-cyclical monotonicity $\sum_{i=1}^n \mathbf{C}_{ii} \leq \sum_{i=1}^n \mathbf{C}_{i\pi(i)}$, so that for the trivial rank-schedule $(r_1) := (n)$ the cost is always $n$-Monge separable.

Given the existence of such a schedule $(r_1, \cdots, r_\kappa)$ with $r_{t+1}$-Monge separability, we proceed by induction on $t \in [0, \kappa]$. For the base case of $t = 0$, as we assume the Monge map exists, for the initial partition $\Gamma_0 = \{(\mathsf{X}, \mathsf{Y})\}$ one has that $\mathsf{Y} = T(\mathsf{X})$. We want to show the variant that $\Gamma_t$ contains sets which are co-clusters of sets with their image under $T$. As the inductive hypothesis, at scale $t > 0$ with $\rho_t$ co-clusters $\Gamma_t = \{(\mathsf{X}_i^{(t)}, \mathsf{Y}_i^{(t)})\}_{i=1}^{\rho_t}$ each satisfies $\mathsf{Y}_i^{(t)} = T(\mathsf{X}_i^{(t)})$. As strict $r_{t+1}$-Monge separability holds for each size $n/\rho_t$ bipartition $(\mathsf{X}_i^{(t)}, \mathsf{Y}_i^{(t)}) \in \Gamma_t$, using Proposition B.4 each such set is divided into $r_{t+1}$ co-clusters $\{(\mathsf{X}_j^{(t+1)}, \mathsf{Y}_j^{(t+1)})\}_{j=1}^{r_{t+1}}$ which satisfy $\mathsf{Y}_j^{(t+1)} = T(\mathsf{X}_j^{(t+1)})$. Thus, taking the union of these $r_{t+1}$ bi-partitions across the $\rho_t$ elements of $\Gamma_t$ we form a set $\Gamma_{t+1}$ of size $\rho_{t+1} = r_{t+1}\rho_t$ which maintains the invariant that $(\mathsf{X}_j^{(t+1)}, \mathsf{Y}_j^{(t+1)}) \in \Gamma_{t+1} \implies \mathsf{Y}_j^{(t+1)} = T(\mathsf{X}_j^{(t+1)})$. At the final level $r_\kappa$ Monge separability holds, so one may conclude on singleton sets of the form $\Gamma_\kappa = \{(x_i, T(x_i))\}_{i=1}^n$. $\qquad \square$

*Remark* B.7. Strict Monge separability applies unconditionally at the terminal level. Observe that all sets in $\Gamma_{\kappa-1}$ have size equal to the rank $(n/\rho_{\kappa-1}) = r_\kappa$, and that we have maintained the invariant that $\mathsf{Y}_j^{(\kappa-1)} = T(\mathsf{X}_j^{(\kappa-1)})$. Let $J_\kappa \subset [n]$ denote the size $r_\kappa$ set of indices for $\mathsf{X}_j^{(\kappa-1)}$ in $\mathsf{X}$. By $c$-cyclical monotonicity, one has for all permutations $\pi \in \mathbf{perm}(n)$

$$\sum_{i=1}^n \mathbf{C}_{ii} = \sum_{i \in J_\kappa} \mathbf{C}_{ii} + \sum_{j \in [n] \setminus J_\kappa} \mathbf{C}_{jj} \leq \sum_{i \in J_\kappa} \mathbf{C}_{i\pi(i)} + \sum_{j \in [n] \setminus J_\kappa} \mathbf{C}_{j\pi(j)} = \sum_{i=1}^n \mathbf{C}_{i\pi(i)}$$

Thus, for the subset of permutations on $n$ where $\pi : \pi \mid_{[n] \setminus J_\kappa} = \mathrm{id}$, we have $\sum_{i \in J_\kappa} \mathbf{C}_{ii} \leq \sum_{i \in J_\kappa} \mathbf{C}_{i\pi(i)}$ implying that one may solve a constant time $O(r_\kappa^2)$ solution to the assignment problem on each size $r_\kappa$ bipartition to recover the final map.

We call $\rho_t$ the effective rank because (to avoid quadratic space complexity) we never instantiate the transport coupling corresponding to the bijective mapping (S12) as a matrix $\mathbf{T}^{(t)}$. Were we to instantiate $\mathbf{T}^{(t)}$, it would have rank $\rho_t$, and moreover we can evaluate its transport cost by using $\mathbf{T}^{(t)}$ to induce a transport coupling $\mathbf{P}^{(t)}$ between the full datasets $\mathsf{X}, \mathsf{Y}$.

$$\mathbf{P}_{ij}^{(t)} := \begin{cases} \rho_t/n^2 & \text{if} \quad q(n/\rho_t) < i, j \leq (q+1)(n/\rho_t) \\ 0 & \text{otherwise} \end{cases}, \tag{S13}$$

where $q \in [\rho_t]$, and where the mass $\rho_t/n^2$ is a simplified form of $(\rho_t/n)^2(1/\rho_t)$. We note that this is a rewriting of $\frac{\rho_t}{n^2} \sum_{q=1}^{\rho_t} \delta_{(\mathbf{x}_i, \mathbf{y}_j) \in \Gamma_{t,q}}$ to have the indices ordered into a contiguous block-structure. Using coupling (S13), which again we never instantiate, one can define:

$$\mathrm{cost}(\mathbf{T}^{(t)}) := \langle \mathbf{C}, \mathbf{P}^{(t)} \rangle.$$

The next proposition shows that the costs $\langle \mathbf{C}, \mathbf{P}^{(t)} \rangle$ decrease as $t$ increases from 1 to $\kappa$, and also provides a bound on their consecutive differences. Below, recall that each $\Gamma_t$ denotes the co-clustering $(\mathsf{X}^{(t)}, \mathsf{Y}^{(t)})$, where

$$\mathsf{X}^{(t)} = \{\mathsf{X}_q^{(t)}\}_{q=1}^{\rho_t}, \quad \mathsf{Y}^{(t)} = \{\mathsf{Y}_q^{(t)}\}_{q=1}^{\rho_t},$$

and where co-cluster $\Gamma_{t,q}$ is defined as:

$$\Gamma_{t,q} := \{(\mathbf{x}, \mathbf{y}) : \mathbf{x} \in \mathsf{X}_q^{(t)}, \mathbf{y} \in \mathsf{Y}_q^{(t)}\}.$$

**Proposition B.8** (Proposition 3.4). *Let cost function $c : \mathbb{R}^d \times \mathbb{R}^d \to \mathbb{R}_+$ be of the form $c(\mathbf{x}, \mathbf{y}) = h(\mathbf{x} - \mathbf{y})$ for some strictly convex function $h : \mathbb{R}^d \to \mathbb{R}_+$ and suppose that $h$ is Lipschitz. Let $\mathbf{P}^{(t)}$ be as defined above in (S13). Then one has the following bound on the difference in cost between iterations of refinement:*

$$0 \leq \langle \mathbf{C}, \mathbf{P}^{(t)} \rangle - \langle \mathbf{C}, \mathbf{P}^{(t+1)} \rangle \leq \|\nabla c\|_\infty \frac{1}{\rho_t} \sum_{q=1}^{\rho_t} \mathrm{diam}(\Gamma_{t,q}), \tag{S14}$$

*where*

$$\mathrm{diam}(\Gamma_{t,q}) \equiv \mathrm{diam}(\mathsf{X}_q^{(t)} \cup T(\mathsf{X}_q^{(t)})) := \max_{\mathbf{x}_i, \mathbf{x}_j, \mathbf{x}_k, \mathbf{x}_l \in \mathsf{X}_q^{(t)}} \left\| (\mathbf{x}_i, T(\mathbf{x}_j)) - (\mathbf{x}_k, T(\mathbf{x}_l)) \right\|.$$

*Proof.* By definition (S13) of $\mathbf{P}^{(t)}$,

$$\langle \mathbf{C}, \mathbf{P}^{(t)} \rangle - \langle \mathbf{C}, \mathbf{P}^{(t+1)} \rangle = \frac{\rho_t}{n^2} \sum_{i=1}^n \sum_{j=1}^n c(\mathbf{x}_i, \mathbf{y}_j) \sum_{q=1}^{\rho_t} \delta_{(\mathbf{x}_i, \mathbf{y}_j) \in \Gamma_{t,q}} - \frac{\rho_{t+1}}{n^2} \sum_{i=1}^n \sum_{j=1}^n c(x_i, y_j) \sum_{q=1}^{\rho_{t+1}} \delta_{(\mathbf{x}_i, \mathbf{y}_j) \in \Gamma_{t+1,q}}$$

$$= \frac{\rho_t}{n^2} \left( \sum_{i=1}^n \sum_{j=1}^n c(\mathbf{x}_i, \mathbf{y}_j) \sum_{q=1}^{\rho_t} \delta_{(\mathbf{x}_i, \mathbf{y}_j) \in \Gamma_{t,q}} - r_{t+1} \sum_{i=1}^n \sum_{j=1}^n c(\mathbf{x}_i, \mathbf{y}_j) \sum_{q'=1}^{\rho_{t+1}} \delta_{(\mathbf{x}_i, \mathbf{y}_j) \in \Gamma_{t+1,q'}} \right)$$

$$= \frac{\rho_t}{n^2} \left( \sum_{q=1}^{\rho_t} \sum_{i=1}^n \sum_{j=1}^n c(\mathbf{x}_i, \mathbf{y}_j) \delta_{(\mathbf{x}_i, \mathbf{y}_j) \in \Gamma_{t,q}} - r_{t+1} \sum_{q'=1}^{\rho_{t+1}} \sum_{i=1}^n \sum_{j=1}^n c(\mathbf{x}_i, \mathbf{y}_j) \delta_{(\mathbf{x}_i, \mathbf{y}_j) \in \Gamma_{t+1,q'}} \right).$$

By Proposition B.4, one then has:

$$= \frac{\rho_{t+1}}{n^2} \left( \sum_{q=1}^{\rho_t} \left( \underbrace{\frac{1}{r_{t+1}} \sum_{i \in \mathsf{X}_q^{(t)}} \sum_{j \in \mathsf{X}_q^{(t)}} c(\mathbf{x}_i, T(\mathbf{x}_j))}_{\text{average ``Monge distortion'' in } \Gamma_{t,q} \text{ over next scale}} - \underbrace{\sum_{z=1}^{r_{t+1}} \sum_{i \in \mathsf{X}_{q\rho_t+z}^{(t+1)}} \sum_{j \in \mathsf{X}_{q\rho_t+z}^{(t+1)}} c(\mathbf{x}_i, T(\mathbf{x}_j))}_{\text{``Monge distortion'' at scale } t+1} \right) \right) \tag{S15}$$

Note that the inner summands of (S15) (indexed by $q$) are non-negative by definition of the refinement step, where *within* each cluster, one has a minimization over a larger set of couplings. This shows $\langle \mathbf{C}, \mathbf{P}^{(t)} \rangle - \langle \mathbf{C}, \mathbf{P}^{(t+1)} \rangle \geq 0$. Towards an upper bound, we will bound each summand of (S15):

$$\left( \frac{1}{r_{t+1}} \sum_{i \in \mathsf{X}_q^{(t)}} \sum_{j \in \mathsf{X}_q^{(t)}} c(\mathbf{x}_i, T(\mathbf{x}_j)) - \sum_{z=1}^{r_{t+1}} \sum_{i \in \mathsf{X}_{q\rho_t+z}^{(t+1)}} \sum_{j \in \mathsf{X}_{q\rho_t+z}^{(t+1)}} c(\mathbf{x}_i, T(\mathbf{x}_j)) \right). \tag{S16}$$

Define $s_{t+1} := n/\rho_{t+1}$ as well as barycenters

$$\bar{\mathbf{x}}^{(t)} := \sum_{\mathbf{x}_i \in \mathsf{X}_{q\rho_t+z}^{(t+1)}} \frac{\mathbf{x}_i}{s_{t+1}}, \quad \bar{\mathbf{y}}^{(t)} := \sum_{\mathbf{x} \in \mathsf{X}_{q\rho_t+z}^{(t+1)}} \frac{T(\mathbf{x}_i)}{s_{t+1}},$$

and note that by Jensen's inequality, for convex cost $c(\cdot, \cdot)$ one has:

$$\sum_{z=1}^{r_{t+1}} \sum_{\mathbf{x}_i \in \mathsf{X}_{q\rho_t+z}^{(t+1)}} \sum_{\mathbf{x}_j \in \mathsf{X}_{q\rho_t+z}^{(t+1)}} c(\mathbf{x}_i, T(\mathbf{x}_j)) = s_{t+1}^2 \sum_{z=1}^{r_{t+1}} \sum_{\mathbf{x}_i \in \mathsf{X}_{q\rho_t+z}^{(t+1)}} \frac{1}{s_{t+1}} \sum_{j \in \mathsf{X}_{q\rho_t+z}^{(t+1)}} \frac{1}{s_{t+1}} c(\mathbf{x}_i, T(\mathbf{x}_j))$$

$$\geq s_{t+1}^2 r_{t+1} c(\bar{\mathbf{x}}^{(t)}, \bar{\mathbf{y}}^{(t)}),$$

so that we may continue upper-bounding the difference (S16):

$$\leq \frac{1}{r_{t+1}} \left( \sum_{\mathbf{x}_i \in \mathsf{X}_q^{(t)}} \sum_{\mathbf{x}_j \in \mathsf{X}_q^{(t)}} c(\mathbf{x}_i, T(\mathbf{x}_j)) \right) - s_{t+1}^2 r_{t+1} c(\bar{\mathbf{x}}^{(t)}, \bar{\mathbf{y}}^{(t)}) \tag{S17}$$

$$= \frac{1}{r_{t+1}} \left( \left( \sum_{\mathbf{x}_i \in \mathsf{X}_q^{(t)}} \sum_{\mathbf{x}_j \in \mathsf{X}_q^{(t)}} c(\mathbf{x}_i, T(\mathbf{x}_j)) \right) - \frac{n^2}{\rho_t} c(\bar{\mathbf{x}}^{(t)}, \bar{\mathbf{y}}^{(t)}) \right) \tag{S18}$$

$$= \frac{1}{r_{t+1}} \left( \sum_{\mathbf{x}_i \in \mathsf{X}_q^{(t)}} \sum_{\mathbf{x}_j \in \mathsf{X}_q^{(t)}} \left( c(\mathbf{x}_i, T(\mathbf{x}_j)) - c(\bar{\mathbf{x}}^{(t)}, \bar{\mathbf{y}}^{(t)}) \right) \right) . \tag{S19}$$

Now, define the diameter of co-cluster $\Gamma_{t,q}$ as follows:

$$\mathrm{diam}\big(\Gamma_{t,q}\big) \equiv \mathrm{diam}\big(\mathsf{X}_q^{(t)} \cup T(\mathsf{X}_q^{(t)})\big) := \max_{\mathbf{x}_i, \mathbf{x}_j, \mathbf{x}_k, \mathbf{x}_l \in \mathsf{X}_q^{(t)}} \left\| \big(\mathbf{x}_i, T(\mathbf{x}_j)\big) - \big(\mathbf{x}_k, T(\mathbf{x}_l)\big) \right\|,$$

Using our Lipschitz assumption on $h$ made at the beginning of the section, where $c(\mathbf{x}, \mathbf{y}) = h(\mathbf{x} - \mathbf{y})$ (we will write $\|\nabla c\|_\infty$ for $\|\nabla h\|_\infty$), one has the inequality:

$$|c(\mathbf{x}_i, T(\mathbf{x}_i)) - c(\mathbf{x}_j, T(\mathbf{x}_j))| \leq \|\nabla c\|_\infty \mathrm{diam}\big(\Gamma_{t,q}\big) .$$

Thus, returning to the bound on each summand (S16), we obtain the upper bound:

$$\leq \frac{1}{r_{t+1}} \sum_{\mathbf{x}_i \in \mathsf{X}_q^{(t)}} \sum_{\mathbf{x}_j \in \mathsf{X}_q^{(t)}} \|\nabla c\|_\infty \left\| \big(\mathbf{x}_i, T(\mathbf{x}_j)\big) - \big(\bar{\mathbf{x}}^{(t)}, \bar{\mathbf{y}}^{(t)}\big) \right\| \tag{S20}$$

As partition $\mathsf{X}^{(t+1)}$ is a refinement of $\mathsf{X}^{(t)}$ and $\mathsf{Y}^{(t+1)}$ is a refinement of $\mathsf{Y}^{(t)}$, it holds that (S16) is upper bounded by:

$$\leq \frac{1}{r_{t+1}} \sum_{i \in \mathsf{X}_q^{(t)}} \sum_{j \in \mathsf{X}_q^{(t)}} \|\nabla c\|_\infty \mathrm{diam}\big(\Gamma_{t,q}\big) , \tag{S21}$$

$$= \frac{1}{r_{t+1}} |\mathsf{X}_q^{(t)}|^2 \|\nabla c\|_\infty \mathrm{diam}\big(\Gamma_{t,q}\big) , \tag{S22}$$

$$= \frac{1}{r_{t+1}} \frac{n^2 \|\nabla c\|_\infty}{\rho_t^2} \mathrm{diam}\big(\Gamma_{t,q}\big) . \tag{S23}$$

To conclude, we plug these bounds into each summand of (S15), obtaining the following bound on the full sum:

$$= \frac{\rho_{t+1}}{n^2} \frac{1}{r_{t+1}} \frac{n^2 \|\nabla c\|_\infty}{\rho_t^2} \sum_{q=1}^{\rho_t} \mathrm{diam}\big(\Gamma_{t,q}\big) \tag{S24}$$

$$= \|\nabla c\|_\infty \frac{1}{\rho_t} \sum_{q=1}^{\rho_t} \mathrm{diam}\big(\Gamma_{t,q}\big). \tag{S25}$$

completing the proof. $\qquad \square$

*Remark* B.9. Proposition B.8 should be considered a *conditional* result. Our proof follows that of (Proposition 1, (Gerber & Maggioni, 2017)), but they are able to provide sharper bounds between elements of a cluster and the centroid of the cluster using the properties assumed to hold in their definition of a multiscale family of partitions (Definition C.3), which mimick the structure of dyadic cubes in Euclidean space. As we do not make any geometric assumptions of our partitions, the above result is a priori weaker, through we leave the exploration of the geometry of partitions induced by low-rank OT to future work.

*Remark* B.10. Note, if $c(\mathbf{x}_i, T(\mathbf{x}_j)) = \gamma$ is constant (i.e., if all points are equidistant in a block), one has that refinement offers no gain from level $\Gamma_t \to \Gamma_{t+1}$:

$$\leq \frac{\rho_{t+1}}{n^2} \sum_{q=1}^{\rho_t} \left| \gamma \frac{|\mathsf{X}_q^{(t)}|^2}{r_{t+1}} - \gamma r_{t+1} |\mathsf{X}_q^{(t+1)}|^2 \right| = \frac{\rho_{t+1}}{n^2} \sum_{q=1}^{\rho_t} \left| \gamma \frac{(n/\rho_t)^2}{r_{t+1}} - \gamma r_{t+1} (n/\rho_{t+1})^2 \right| = 0 \,.$$

*Remark* B.11. The work (Seguy et al., 2018) suggests a loss dependent on an (entropic) coupling $\gamma$. If $\gamma$ is sparse and supported on the graph of the Monge map so that $\gamma = (\mathrm{id} \times T)_\sharp \, \mu$, this loss becomes a regression of a neural network $T_\theta$ on the Monge map $T$ over the support of $\mu$: $\min_{T_\theta} \mathbb{E}_\mu c \, (T_\theta(\mathbf{x}_i), T(\mathbf{x}_i))$.

*Proof.* By linearity of the push-forward map one immediately obtains

$$\int_{\mathsf{X}\times\mathsf{Y}} \|T_\theta(x) - y\|_2^p (\mathrm{id} \times T)_\sharp \sum_{i=1}^n \mu_i \delta_{x_i} \mathrm{d}x \mathrm{d}y = \int_{\mathsf{X}\times\mathsf{Y}} \|T_\theta(x) - y\|_2^p \sum_{i=1}^n \mu_i (\mathrm{id} \times T)_\sharp \delta_{x_i} \mathrm{d}x \mathrm{d}y$$

$$= \sum_{i=1}^n \mu_i \int_{\mathsf{X}\times\mathsf{Y}} \|T_\theta(x) - y\|_2^p \delta_{(x_i, T(x_i))} \mathrm{d}y \mathrm{d}x = \sum_{i=1}^n \mu_i \|T_\theta(x_i) - T(x_i)\|_2^p \,,$$

By integrating against the $\delta$. As $\mu_i > 0$, it holds that this loss is identically zero if and only if $T_\theta = T$ on the dataset $(x_i)_{i=1}^n$

$$\min_{T_\theta} \int_{\mathsf{X}\times\mathsf{Y}} \|T_\theta(x) - y\|_2^p d\gamma(x,y) = 0 \iff \|T_\theta(x_i) - T(x_i)\|_2^p = 0 \iff T_\theta(x_i) = T(x_i)$$

$\square$

In other words, when one minimizes the objective of (Seguy et al., 2018) using the bijective Monge map $\gamma = (\mathrm{id} \times T)_\sharp \mu$ as opposed to an entropic coupling, the objective of (Seguy et al., 2018) reduces to an unbiased regression. That is, the neural map $T_\theta$ directly matches $T$ over the dataset support as if trained on supervised $(x, y)$ pairs $y = T(x)$.

## C. Background: Multiscale Optimal Transport

### C.1. Multiscale Partitions

(Gerber & Maggioni, 2017) describe a general multiscale strategy for computing OT couplings between metric measure spaces $(\mathsf{X}, \mathsf{d}_\mathsf{X}, \mu)$ and $(\mathsf{Y}, \mathsf{d}_\mathsf{Y}, \nu)$. They state this in the Kantorovich setting, using a general cost function $c : \mathsf{X} \times \mathsf{Y} \to \mathbb{R}_+$. Their framework consists of several elements:

1. A way of *coarsening* the set of source points $\mathsf{X}$ and the measure $\mu$ across multiple scales:

$$(\mathsf{X}, \mu) =: (\mathsf{X}_J, \mu_J) \to (\mathsf{X}_{J-1}, \mu_{J-1}) \to \cdots \to (\mathsf{X}_1, \mu_1) \,, \tag{S26}$$

   as well as an analogous coarsening for the set of target points $\mathsf{Y}$:

$$(\mathsf{Y}, \nu) =: (\mathsf{Y}_J, \nu_J) \to (\mathsf{Y}_{J-1}, \nu_{J-1}) \to \cdots \to (\mathsf{Y}_1, \nu_1) \,, \tag{S27}$$

   where at each scale $j$, $\mathrm{supp}(\mu_j) = \mathsf{X}_j$ and $\mathrm{supp}(\nu_j) = \mathsf{Y}_j$, and the cardinality of each $\mathsf{X}_j$ and $\mathsf{Y}_j$ decreases with $j$.

2. A way of *propagating* coupling $\pi_j$ solving the transport problem $\mu_j \to \nu_j$ at scale $j$ to a coupling $\pi_{j+1}$ at scale $j+1$.

3. A way of *refining the coupling* from scale $j$ to an optimal solution at scale $j+1$.

To derive approximation bounds for the error incurred by the multiscale transport problem at each scale, (Gerber & Maggioni, 2017) use regular families of multiscale partitions (Definition C.3 below) to define approximations to $\mu, \nu$ and c at all scales.

For $z \in X$, define $B_x(r) := \{x' \in X : d_X(x, x') < r\}$ as the metric ball of radius $r$ centered at $x$. Functions $f, g : X \to \mathbb{R}$ have the *same order of magnitude* if there is $c_1, c_2 > 0$ with $c_1 f(x) \leq g(x) \leq c_2 f(x)$ for all $x \in X$, and in this case we write $f \asymp g$. Write $\mathcal{M}(X)$ for the space of unsigned measures on $X$, and write $\mathcal{P}(X)$ for the subspace of probability measures.

**Definition C.1.** A metric space $(X, d_X)$ has *doubling dimension* $d > 0$ if every $B_z(r)$ admits a covering by at most $2^d$ balls of radius $r/2$.

A metric space is said to be *doubling* if it has doubling dimension $d$ for some $d > 0$. A related notion to a doubling metric space is a doubling measure.

**Definition C.2.** Measure $\mu \in \mathcal{M}(X)$ is a *doubling measure with dimension* $d$ if there is a constant $c_1 > 0$ such that for all $x \in X$ and all $r > 0$, one has $c_1^{-1} r^d \leq \mu(B_x(r)) \leq c_1 r^d$, i.e. $\mu(B_x(r)) \asymp r^d$.

Note that if $(X, d_X, \mu)$ is doubling, then $d_X$ is doubling, and up to modification of $d_X$ to an equivalent metric, the dimension $d$ can be taken as the same in either case.

**Definition C.3.** Given metric measure space $(X, d_X, \mu)$, a *regular family of multiscale partitions* with scaling parameter $\theta > 1$ is a family of sets

$$\left\{ \{C_{j,k}\}_{k=1}^{K_j} \right\}_{j=1}^{J},$$

with each $C_{j,k} \subset X$ such that:

1. For each scale $j$, the sets $\{C_{j,k}\}_{k=1}^{K_j}$ partition $X$.

2. For each scale $j \in [J-1]$, either $C_{j+1,k'} \cap C_{j,k} = \emptyset$ or $C_{j+1,k'} \subset C_{j,k}$. In this latter case, we say that $(j+1, k')$ is a *child* of $(j, k)$, or equivalently that $(j, k)$ is a *parent* of $(j+1, k')$, writing $(j+1, k') \prec (j, k)$.

3. There is a constant $A > 0$ such that for all $j, k$, we have diameter $\operatorname{diam}(C_{j,k}) \leq A\theta^{-j}$.

4. Each $C_{j,k}$ contains a "center point" $c_{j,k}$ such that $B_{c_{j,k}}(\theta^{-j}) \subset C_{j,k}$.

We take $\theta = 2$ for simplicity. As the child-parent terminology suggests, these partitions (through the second point) have a tree structure, like dyadic cubes in $\mathbb{R}^d$. Though the measure $\mu$ is not explicitly used in the above definition, the third and fourth points imply $\mu(C_{j,k}) \asymp 2^{-jd}$ and $K_j \asymp 2^{jd}$.

**Coarsening spaces and measures**   Now suppose that each of $X$ and $Y$ are each discrete metric measure spaces, each equipped with regular families $\Gamma(X), \Gamma(Y)$ of multiscale partitions:

$$\Gamma(X) := \{\Gamma_j(X)\}_{j=0}^J, \quad \Gamma_j(X) := \{C_{j,k}(X)\}_{k=1}^{K_j(X)}$$
$$\Gamma(Y) := \{\Gamma_j(Y)\}_{j=0}^J, \quad \Gamma_j(Y) := \{C_{j,k}(Y)\}_{k=1}^{K_j(Y)},$$

and these yield the coarsening chains in (S26), (S27) in the most natural way possible at each scale $j$, defining the coarse-grained spaces $X_j, Y_j$ to be the clusters at scale $j$:

$$X_j := \Gamma_j(X), \quad Y_j := \Gamma_j(Y),$$

while the measures at scale $j$ are defined from the measures at scale $j+1$ via:

$$\mu_j(C_{j,k}(X)) := \sum_{(j+1,k') \prec (j,k)} \mu_{j+1}(C_{j+1,k'}(X)), \quad \nu_j(C_{j,k}(Y)) := \sum_{(j+1,k') \prec (j,k)} \nu_{j+1}(C_{j+1,k'}(Y)).$$

The fourth item of Definition C.3 requires that we define cluster centers $\bar{c}_{j,k}(X)$ for each $C_{j,k}(X)$. At the finest scale $j = J$, all clusters $C_{J,k}(X)$ correspond to singletons $\{x_{J,k}\}$, so we define $\bar{c}_{J,k}(X) := x_{J,k}$ in this case. At coarser scales, these centers can be defined recursively from the next finest scale, depending on the structure of $X$.

For example, if $\mathsf{X}$ has vector space structure (in addition to being a metric measure space), a natural choice for cluster centers $x_{j,k}$ at scale $j = 0, \ldots, J-1$ is the weighted average $x_{j,k} := \bar{c}_{j,k}(\mathsf{X})$, where

$$\bar{c}_{j,k}(\mathsf{X}) := \sum_{(j+1,k') \prec (j,k)} \mu_{j+1}(C_{j+1,k'}(\mathsf{X})) x_{j+1,k'} \, .$$

On the other hand, in the absence of vector space structure, one can still define

$$\bar{c}_{j,k}(\mathsf{X}) = \arg\min_{x \in \mathsf{X}} \sum_{(j+1,k') \prec (j,k)} \mathsf{d}_\mathsf{X}^p(x, x_{j+1,k'}) \, ,$$

with analogous constructions for $\mathsf{Y}$ yielding centers $y_{j,k}$.

**Coarsening the cost function**  (Gerber & Maggioni, 2017) suggest three ways to coarsen the cost function using the multiscale partition. To condense the notation slightly, let us write $x_{j,k}$ in place of $C_{j,k}(\mathsf{X})$ and $y_{j,k'}$ in place of $C_{j,k}(\mathsf{X})$ and $C_{j,k'}(\mathsf{Y})$.

(c-i) The pointwise value

$$\mathsf{c}_j(\,x_{j,k},\, y_{j,k'}\,) := \mathsf{c}(x_{j,k}, y_{j,k'}) \, , \tag{S28}$$

using centers $x_{j,k}$ and $y_{j,k'}$ defined in any of the ways above.

(c-ii) The local average

$$\mathsf{c}_j(\,x_{j,k},\, y_{j,k'}\,) := \frac{\sum_{x \in C_{j,k}(\mathsf{X}), y \in C_{j,k'}(\mathsf{Y})} \mathsf{c}(x,y)}{|C_{j,k}(\mathsf{X})|\,|C_{j,k'}(\mathsf{Y})|}$$

(c-iii) The local weighted average:

$$\mathsf{c}_j(\,x_{j,k},\, y_{j,k'}\,) := \frac{\sum_{x \in C_{j,k}(\mathsf{X}), y \in C_{j,k'}(\mathsf{Y})} \mathsf{c}(x,y) \pi_{j-1}^\star(x_{j-1,k_1}, y_{j-1,k_1'})}{\sum_{x \in C_{j,k}(\mathsf{X}), y \in C_{j,k'}(\mathsf{Y})} \pi_{j-1}^\star(x_{j-1,k_1}, y_{j-1,k_1'})} \, ,$$

where $\pi_{j-1}^\star$ is the optimal (or approximately optimal) OT coupling at scale $j-1$, defined below. The indices $k_1$ and $k_1'$ are defined using the tree structure of the partition: $k_1$ is the unique index among $[K_{j-1}(\mathsf{X})]$ such that $(j,k) \prec (j-1, k_1)$, and likewise $k_1'$ is unique among $[K_{j-1}(\mathsf{X})]$.

## C.2. Propagation of OT solutions across scales

For each scale $j$, consider the OT problem given as follows.

$$\pi_j^\star := \arg\min_{\pi \in \Pi(\mu_j, \nu_j)} \mathrm{cost}(\pi_j), \quad \text{where:}$$

$$\mathrm{cost}(\pi_j) := \sum_{k \in [K_j(\mathsf{X})],\, k' \in [K_j(\mathsf{Y})]} c(x_{j,k}, y_{j,k'}) \pi_j(x_{j,k}, y_{j,k'}) \tag{S29}$$

(Gerber & Maggioni, 2017) show bounds on $|\mathrm{cost}(\pi_j^\star) - \mathrm{cost}(\pi_j^\star)|$ of a constant times $2^{-j} \|\nabla \mathsf{c}\|_\infty$, but note that this only implies closeness of the couplings in terms of their cost, not necessarily in any other sense.

Given an optimal coupling $\pi_j^\star$ at scale $j$, (Glimm & Henscheid, 2013) proposed a direct propagation strategy to initialize the problem at scale $j+1$, distributing the mass $\pi_j^\star(x_{j,k}, y_{j,k'})$ equally to all combinations of paths between $\mathrm{children}(x_{j,k})$ and $\mathrm{children}(y_{j,k'})$. In this context, a path is understood to mean a source-target pair at the next scale, e.g. a pair of the form $(x_{j+1,\ell}, y_{j+1,\ell'})$. To formalize this, let

$$\mathcal{A}_j := \{(x_{j,k}, y_{j,k'}) : k \in [K_j(\mathsf{X})],\, k' \in [K_j(\mathsf{Y})]\}$$

denote *all* paths between points in $\mathsf{X}_j$ and $\mathsf{Y}_j$. The drawback of this warm-start procedure is that if $\mathrm{supp}(\mu_j) \subset \mathcal{A}_j$, which is always the case, the refinement procedure still requires quadratic space complexity at the finest scale.

To mitigate the ultimate quadratic space complexity of retaining all possible paths at all scales, (Gerber & Maggioni, 2017) allow for a refinement procedure where the support of couplings at scale $j + 1$ is restricted to a subset $\mathcal{R}_{j+1} \subset \mathcal{A}_{j+1}$ of all possible paths (with $\mathcal{R}_{j+1}$ defined by the optimal coupling at the previous iteration). Given $\mathcal{R}_j \subset \mathcal{A}_j$, let $\pi_j^\star|_{\mathcal{R}_j}$ denote the optimal solution to the path-restricted or *restricted problem* at scale $j$:

$$\pi_j^\star|_{\mathcal{R}_j} := \underset{\substack{\pi \in \Pi(\mu_j, \nu_j), \\ \mathrm{supp}(\pi) \subset \mathcal{R}_j}}{\arg\min} \; \mathrm{cost}(\pi_j) \,. \tag{S30}$$

**Simple propagation.**    The simplest way to restrict the number of paths considered at subsequent scales is to use paths at scale $j$ whose endpoints are children of mass-bearing paths at scale $j + 1$:

$$\mathcal{R}_{j+1} := \{(x_{j+1,\ell}, y_{j+1,\ell}) \colon \exists (x_{j,k}, y_{j,k'}) \in \mathrm{supp}(\pi_j^\star) \text{ s.t } (j+1, \ell) \prec (j, k) \text{ and } (j+1, \ell') \prec (j, k')\} \,.$$

The optimal Kantorovich plan at scale $j$ has at most $(K_j(\mathsf{X}) + K_j(\mathsf{Y}) + 1)$ non-zero entries. Using the above simple propagation strategy constrains plan at scale $j + 1$ to be supported on at most

$$\alpha_j^2 (K_j(\mathsf{X}) + K_j(\mathsf{Y}))$$

entries, where $\alpha_j$ is the maximum number of children of any $(j, k)$ across both datasets. When the ambient space has doubling dimension $d$, for any $j$ one has $\alpha_j \asymp 2^d$, yielding a plan with *linear* space complexity at the finest scale.

**Capacity constraint propagation.**    This propagation strategy solves a modified minimum flow problem at scale $j$ in order to include additional paths at scale $j + 1$ likely to be included in the optimal solution $\pi_{j+1}^\star$. Concretely, one first computes an unconstrained optimal plan $\pi_j^\star|_{\mathcal{R}_j}$ at scale $j$. Then, a new OT plan $\tilde{\pi}_j^\star|_{\mathcal{R}_j}$ is solved for at scale $j$ now subject to the capacity constraint

$$\tilde{\pi}_j^\star|_{\mathcal{R}_j} \le U_{k,k'} \min(\mu(x_{j,k}), \nu(y_{j,k'}))$$

for each $(x_{j,k}, y_{j,k'}) \in \mathrm{supp}(\pi_j^\star|_{\mathcal{R}_j})$, where the random variables $U_{k,k'}$ are i.i.d. $\mathrm{Uniform}([0.1, 0.9])$. This can also be iterated several times, in all cases leading to linear space complexity in the optimization at the finest scale.

### C.3. Refinement of the propagated solution

Propagation of a solution to the restricted transport problem (S30) at scale $j$, in general cannot guarantee reaching an optimal solution to the restricted problem at scale $j + 1$, and can lead to accumulation of errors across all scales. Several *refinement* strategies are proposed in (Gerber & Maggioni, 2017) to address this.

**Potential Refinement.**    One refinement strategy leverages the problem dual to (3), here stated at the finest scale:

$$\max_{\substack{\mathbf{f} \in \mathbb{R}^n, \mathbf{g} \in \mathbb{R}^m \\ \mathbf{f}_i + \mathbf{g}_j \le \mathbf{C}_{ij}}} \sum_{i=1}^n \mu(\{x_i\}) \mathbf{f}_i + \sum_{j=1}^m \mu(\{y_j\}) \mathbf{g}_j \,. \tag{S31}$$

The refinement strategy uses optimal dual variables $\mathbf{f}^\star, \mathbf{g}^\star$ to select paths to include at the next scale. From the dual formulation, an optimal solution $(\mathbf{f}^\star, \mathbf{g}^\star)$ to (S31) must have all nonnegative entries in the *reduced cost matrix*, defined as the matrix $\mathbf{C} - \mathbf{f} \oplus \mathbf{g}$ with entries $\mathbf{C}_{kk'} - \mathbf{f}_k - \mathbf{g}_{k'}$. Note that the dual to the restricted problem (S30) is well-defined, and for this dual we denote the optimal dual potentials by $\mathbf{f}^\star|_{\mathcal{R}_j}$ and $\mathbf{g}^\star|_{\mathcal{R}_j}$. With slight abuse of notation, let $(\mathbf{f}^\star \oplus \mathbf{g}^\star)|_{\mathcal{R}_j}$ be

$$(\mathbf{f}^\star \oplus \mathbf{g}^\star)|_{\mathcal{R}_j} := (\mathbf{f}^\star|_{\mathcal{R}_j} \oplus \mathbf{g}^\star|_{\mathcal{R}_j}) \odot \mathbf{M}^{(j)} \,,$$

where $\mathbf{M}^{(j)}$ is the indicator matrix of the restricted set of paths $\mathcal{R}_j$ at scale $j$, and where $\odot$ denotes the Hadamard (entrywise) product. While the restricted set of paths $\mathcal{R}_j$ is inherited from the previous scale, one can define a new set of paths $\mathcal{V}_j^0$ based on where the restricted reduced cost $\mathbf{C} - (\mathbf{f}^\star \oplus \mathbf{g}^\star)|_{\mathcal{R}_j}$ is nonpositive:

$$\mathcal{V}_j^0(\pi_j^\star|_{\mathcal{R}_j}) := \{(x_{j,k}, y_{j,k'}) \in \mathcal{A}_j : \mathbf{C}_{kk'} - [(\mathbf{f}^\star \oplus \mathbf{g}^\star)|_{\mathcal{R}_j}]_{kk'} \le 0\} \,.$$

*Table S1.* Hyperparameters for Synthetic Experiments

| Parameter Name | Variable | Value |
|---|---|---|
| Rank-Annealing Schedule | $(r_1, \ldots, r_\kappa)$ | [2, 512] |
| Hierarchy Depth | $\kappa$ | 2 |
| Maximal Base Rank | $Q$ | $2^{10}$ |
| Maximal Intermediate Rank | $C$ | 16 |

With a new set of paths $\mathcal{Q}_j^0 := \mathcal{V}_j^0(\pi_j^\star|_{\mathcal{R}_j})$, one can compute a new optimal plan $\pi_j^\star|_{\mathcal{Q}_j^0}$ at scale $j$ restricted to these paths, as well as *new* optimal dual potentials $\mathbf{f}^\star|_{\mathcal{V}_j^0}$ and $\mathbf{g}^\star|_{\mathcal{V}_j^0}$ leading to a new reduced cost $\mathbf{C} - (\mathbf{f}^\star \oplus \mathbf{g}^\star)|_{\mathcal{V}_j^0}$. This strategy can be iterated via

$$\mathcal{Q}_j^i := \mathcal{V}_j(\pi_j^\star|_{\mathcal{Q}_j^{i-1}}), \tag{S32}$$

yielding the sequence of transport plans $\pi_j^\star|_{\mathcal{Q}_j^i}$, all at scale $j$, which converge on a solution whose reduced cost is nonnegative, necessarily making it optimal. The potential refinement strategy was used by (Glimm & Henscheid, 2013), with (Schmitzer, 2016) introducing shielding neighborhoods in a similar spirit, using dual potentials to locally verify global optimality.

## D. Experimental Details

### D.1. Synthetic Experiments

For all of the synthetic experiments, we first generate $n = 1024$ points from three datasets: the checkerboard dataset ((Makkuva et al., 2020)), the MAFMoons and Rings dataset ((Buzun et al., 2024)), and the Half-moon and S-curve dataset ((Buzun et al., 2024)). Following (Buzun et al., 2024) the random seed was set to 0 for data-generation with `jax.random.key(0)`. We evaluate the OT cost $\langle \mathbf{C}, \mathbf{P} \rangle_F$ of `HiRef` Sinkhorn (Cuturi, 2013), and ProgOT (Kassraie et al., 2024) on each of these three datasets, where we use (1) the Euclidean cost $\|\cdot\|_2$, and (2) the squared Euclidean cost $\|\cdot\|_2^2$ (Table S2). We additionally quantify the number of non-zero entries in the plan and its entropy (Table S3).

We also compare the cost of couplings computed by Hierarchical Refinement to low-rank couplings (Scetbon et al., 2021; Halmos et al., 2024) of varying rank. We observe that as the latent rank $r \to n$, the OT cost $\langle \mathbf{C}, \mathbf{P}_r \rangle_F$ asymptotically approaches the cost achieved by Hierarchical Refinement (Figure S3). In the limit $\lim_{r \to n} \langle \mathbf{C}, \mathbf{P}_r \rangle_F$ low-rank OT recovers Sinkhorn (Scetbon & Cuturi, 2022) and approaches quadratic memory complexity, while `HiRef` remains linear in space.

### Checkerboard

The checkerboard dataset (Makkuva et al., 2020) is defined by random variables $Y \sim Q$ sampled from the source distribution according to $\mathbf{Y} = \mathbf{X} + \mathbf{Z}$ where $\mathbf{X}$ and $\mathbf{Z}$ are sampled from Uniform distributions defined by

$$\mathbf{X} \sim \text{Uniform}\left(\{(0,0), (1,1), (1,-1), (-1,1), (-1,-1)\}\right),$$
$$\mathbf{Z} \sim \text{Uniform}\left([-0.5, 0.5] \times [-0.5, 0.5]\right).$$

the target distribution $P$ has random variable $\mathbf{Y}'$ where the random variable $\mathbf{Y}'$ is defined as $\mathbf{Y}' = \mathbf{X}' + \mathbf{Z}$ with components

$$\mathbf{X}' \sim \text{Uniform}\left(\{(0,1), (0,-1), (1,0), (-1,0)\}\right),$$
$$\mathbf{Z} \sim \text{Uniform}\left([-0.5, 0.5] \times [-0.5, 0.5]\right).$$

### MAFMoons and Rings

The MAFMoon dataset (Buzun et al., 2024) defines a source distribution $Q$ by sampling $\mathbf{X} \sim \mathcal{N}(0, \mathbb{1}_2)$ and applying the non-linear transformation defined by

$$\mathbf{Y} = \begin{bmatrix} Y_1 \\ Y_2 \end{bmatrix} = \begin{bmatrix} 0.5(X_1 + X_2^2) - 5 \\ X_2 \end{bmatrix}$$

this introduces a quadratic dependency on the Gaussian randomly variable to generate a crescent shape.

The target distribution $P$ representing concentric rings is generated by first sampling $\theta \sim \text{Uniform}(2\pi)$, with fixed radii $r_i \in \{0.25, 0.55, 0.9, 1.2\}$ from which one transforms to Cartesian coordinates as $x_i = 3r_i \cos\theta_i$ and $y_i = 3r_i \sin\theta_i$. Gaussian noise is added to each of these, as $\epsilon \sim \mathcal{N}(0, \mathbb{1}\sigma^2)$ for $\sigma = 0.08$.

**Half-moon and S-Curve**

The Half-moon and S-curve dataset (Buzun et al., 2024) is generated from $\mathbf{Y} = \texttt{make\_moons}$ and $\texttt{make\_S\_curve}$ from the $\texttt{scikit-learn}$ library. Both datasets are transformed further with a rotation $\mathbf{R}(\theta)$, a scaling $\lambda$, and a translation $\mu$ applied as $\mathbf{Y}' \leftarrow \mathbf{R}(\theta)(\lambda\mathbf{Y}) + \mu$.

*Table S2.* Comparison Table for Coupling-Based OT Methods on Primal Cost $\langle \mathbf{C}, \mathbf{P} \rangle_F$ for $\|\cdot\|_2$ and $\|\cdot\|_2^2$

| Method | Checkerboard (Makkuva 2020) | | MAFMoons & Rings (Buzun 2024) | | Half Moon & S-Curve (Buzun 2024) | |
|---|---|---|---|---|---|---|
| | $\|\cdot\|_2$ | $\|\cdot\|_2^2$ | $\|\cdot\|_2$ | $\|\cdot\|_2^2$ | $\|\cdot\|_2$ | $\|\cdot\|_2^2$ |
| Sinkhorn | 0.3573 | 0.1319 | 0.4422 | 0.4440 | **0.5663** | **0.5663** |
| ProgOT | N/A | 0.1320 | N/A | 0.4443 | N/A | 0.5709 |
| HiRef | **0.3533** | **0.1248** | **0.4398** | **0.4414** | 0.5741 | 0.5737 |

*Table S3.* Entropy and Non-Zero Entries ($> 10^{-8}$) of Coupling Matrices for Each Method and Dataset (Wasserstein-2 distance cost, $\|\cdot\|_2^2$)

| Method | Checkerboard (Makkuva 2020) | | MAFMoons & Rings (Buzun 2024) | | Half Moon & S-Curve (Buzun 2024) | |
|---|---|---|---|---|---|---|
| | Entropy | Non-Zeros | Entropy | Non-Zeros | Entropy | Non-Zeros |
| Sinkhorn | 12.8509 | 624733 | 12.6117 | 678720 | 12.7776 | 652993 |
| ProgOT | 12.3830 | 271087 | 11.6158 | 327764 | 12.1170 | 337258 |
| HiRef | **6.9314** | **1024** | **6.9314** | **1024** | **6.9314** | **1024** |

*Table S4.* Comparison of Coupling-Based OT Methods on Primal Cost $\langle \mathbf{C}, \mathbf{P} \rangle_F$ (Wasserstein-2) on 512 point small instance.

| Method | Checkerboard | MAF Moons & Rings | Half Moon & S-Curve |
|---|---|---|---|
| MOP (Gerber & Maggioni, 2017) | 0.393 | 0.276 | 0.401 |
| Sinkhorn ($\texttt{ott-jax}$) | 0.136 | 0.221 | 0.338 |
| ProgOT | 0.136 | 0.216 | 0.334 |
| HiRef | 0.129 | 0.216 | 0.334 |
| Dual Revised Simplex Solver | **0.127** | **0.214** | **0.332** |

*Table S5.* Hyperparameters for Mouse-Embryo Spatial Transcriptomics Experiment (E15.5-16.5)

| Parameter Name | Variable | Value |
|---|---|---|
| Rank-Annealing Schedule | $(r_1, \ldots, r_\kappa)$ | [2, 86, 659] |
| Hierarchy Depth | $\kappa$ | 3 |
| Maximal Base Rank | $Q$ | $2^{10}$ |
| Maximal Intermediate Rank | $C$ | 128 |

## D.2. Large-scale Transcriptomics Matching on Mouse-Embryo

In this problem, we use `HiRef` to find a full-rank alignment matrix between successive pairs of spatial transcriptomics (ST) (Ståhl et al., 2016) slices. These are from a dataset of whole-mouse embryogenesis (Chen et al., 2022) on the Stereo-Seq platform. These datasets have been measured at successive 1-day time-intervals: E9.5 ($n = 5913$), E10.5 ($n = 18408$), E11.5 ($n = 30124$), E12.5 ($n = 51365$), E13.5 ($n = 77369$), E14.5 ($n = 102519$), E15.5 ($n = 113350$), and E16.5 ($n = 121767$), where the embryonic mouse is growing across the stages so that the sample-complexity $n$ increases with the numeric stage. For each pair of datasets of size $n$ and $m$, we sub-sample the datasets so that the size of the two datasets is given as $n \leftarrow \min\{n, m\}$.

In the context of spatial transcriptomics, an experiment conducted on a two-dimensional tissue slice produces a data pair $(\mathbf{X}, \mathbf{Z})$. Here, $\mathbf{X} \in \mathbb{R}^{n \times p}$ represents the gene expression matrix, where n denotes the number of cells (or spatial spots) analyzed on the slice, and p signifies the number of genes measured. Specifically, the entry $\mathbf{X}_{ij} \in \mathbb{R}_+$ corresponds to the expression level of gene j in cell i, with higher values indicating greater expression intensity. Concurrently, $\mathbf{Z} \in \mathbb{R}^{n \times 2}$ is the spatial coordinate matrix, where each row i contains the (x, y) coordinates of cell i on the tissue slice. Consequently, every cell on the slice is characterized by a gene expression vector of length p, capturing its molecular features, and a coordinate vector of length two, detailing its spatial position within the slice.

We utilize the extensive, real-world dataset on mouse embryo development presented in (Chen et al., 2022), which encompasses eight temporal snapshots of spatial transcriptomics (ST) slices throughout the entire mouse embryo development process. And align all consecutive timepoints. The preprocessing of this dataset is conducted using the standard SCANPY (Wolf et al., 2018) workflow. Initially, we ensure that both slices contain an identical set of genes by filtering, which results in a common gene set across all cells for each pair of timepoints. Subsequently, we apply log-normalization to the gene expression data of all cells from the two slices. To compress the data, we perform Principal Component Analysis (PCA), reducing the dimensionality of the gene expression profiles to $d = 60$ PCs. Finally, we compute the Euclidean distances between gene expression vectors in the PCA-transformed space to construct the cost matrix $\mathbf{C}$, on which we solve a Wasserstein problem to obtain the optimal coupling $\mathbf{P}$ of full-rank. We offer hyperparameters for the E15-16.5 experiment (the largest alignment) in Table S5. For the other experiments, the maximal intermediate rank is $r = 16$ up to E10.5, $r = 32$ to E11.5, $r = 64$ up to E13.5, and 128 for E14.5-16.5. The rank-annealing schedule is generated according to the dynamic program in each case by the `rank_annealing.optimal_rank_schedule( n, hierarchy_depth , max_Q , max_rank )` function.

In this experiment, we benchmark against the default implementation of Sinkhorn in `ott-jax` (Cuturi et al., 2022) with entropy parameter $\epsilon = 0.05$, and additionally benchmark against the default implementations of `ProgOT` (Kassraie et al., 2024) and `LOT` (Scetbon et al., 2021) in `ott-jax`. For the low-rank methods `LOT` and `FRLC` (Halmos et al., 2024) we fix a constant rank of $r = 40$ for these experiments. While LOT (Scetbon et al., 2021) provides a robust, scalable low-rank procedure for the Wasserstein-2 distance, the LOT solver with point cloud input on Wasserstein-1 cost only runs for the first pair (E9.5:E10.5). For subsequent pairs we input the cost $\mathbf{C}$ directly, resulting in the LOT solver running up to the third pair (E11.5:E12.5). Mini-batch OT is run with batch-sizes ranging from 128 to 2048, and is performed without replacement. As noted in prior works (Fatras et al., 2020; 2021a;b), this is a standard choice for instantiating a full-rank coupling using mini-batch OT. Sinkhorn is used to solve each mini-batch coupling, as implemented in `ott-jax` with a default setting of the entropy parameter $\epsilon = 0.05$.

## D.3. Brain Atlas Spatial Alignment

We took inspiration from MERFISH-MERFISH alignment experiments of (Clifton et al., 2023), particularly gene abundance transfer tasks that STalign is exhibited on. The data are available on the Vizgen website for MERFISH Mouse Brain Receptor Map data release (https://info.vizgen.com/mouse-brain-map). The two spatial transcriptomics slices used for the experiment

*Table S6.* Cost Values $\langle \mathbf{P}, \mathbf{C} \rangle_F$ for Different Methods Across Embryonic Stages

| Method | E9.5-E10.5 | E10.5-E11.5 | E11.5-E12.5 | E12.5-E13.5 | E13.5-E14.5 | E14.5-E15.5 | E15.5-E16.5 |
|--------|------------|-------------|-------------|-------------|-------------|-------------|-------------|
| HiRef | **21.81** | **14.81** | **16.14** | **14.35** | **13.78** | **14.29** | **12.79** |
| Sinkhorn | 21.91 | 14.89 | - | - | - | - | - |
| ProgOT | 22.56 | 15.35 | - | - | - | - | - |
| MB 128 | 22.44 | 15.35 | 16.69 | 14.86 | 14.14 | 14.75 | 13.32 |
| MB 512 | 22.15 | 15.05 | 16.33 | 14.54 | 13.92 | 14.50 | 13.01 |
| MB 1024 | 22.05 | 15.02 | 16.24 | 14.45 | 13.86 | 14.43 | 12.91 |
| MB 2048 | 21.98 | 14.98 | 16.18 | 14.39 | 13.81 | 14.39 | 12.85 |
| FRLC | 23.14 | 16.09 | 17.74 | 15.47 | 14.64 | 15.51 | 14.00 |
| LOT | 26.06 | 19.06 | 21.64 | - | - | - | - |

are slice 2, replicate 3 ("source" dataset) and slice 2, replicate 2 ("target" dataset). The datasets will be denoted $(\mathbf{X}^1, \mathbf{S}^1)$ for the source and $(\mathbf{X}^2, \mathbf{S}^2)$ for the target.

The source dataset consists of $85,958$ spots, while the target dataset consists of $84,172$ spots. To apply HiRef to these data, we subsampled the source dataset to have $84,172$ spots also (uniformly at random), removing a total of 1786 spots. We use this sub-sampled $n \times n$ dataset for all methods, but as discussed below, note that this sub-sampling incurs little error. We ran HiRef using the settings max_rank = 11 and hierarchy_depth=4, for a total runtime of 10 minutes 6 seconds, on an A100 GPU. The random seed was set to 44. For the cost function used by HiRef, we only use the *spatial* modalities $\mathbf{S}^1, \mathbf{S}^2$ of the two datasets. We centered the spatial coordinates of both datasets, and applied a rotation by 45 degrees to the first dataset. With these registered spatial data, here denoted $\tilde{\mathbf{S}}^1 = \{\mathbf{s}_i^1\}_{i=1}^n$ and $\tilde{\mathbf{S}}^2 = \{\mathbf{s}_i^2\}_{i=1}^n$, we formed the cost matrix $\mathbf{C}$ given by:

$$\mathbf{C}_{ij} = \|\mathbf{s}_i^1 - \mathbf{s}_j^2\|_2,$$

where $\| \cdot \|_2$ denotes the Euclidean distance between the spatial coordinates. This cost $\mathbf{C}$ was used as input to HiRef, which produced as output a 1-1 mapping $T$ between the two datasets (a permutation matrix is too large to instantiate).

We then evaluated the performance of HiRef through cosine similarity of predicted gene abundance with target gene abundance, across five "spatially-patterned genes" (using the terminology of (Clifton et al., 2023)): *Slc17a7, Grm4, Olig1, Gad1, Peg10*. Writing $\mathfrak{g}$ to stand in for any of these genes, we formed the abundance vectors $\mathbf{v}^{1,\mathfrak{g}}$ and $\mathbf{v}^{2,\mathfrak{g}}$ using the raw counts for gene $\mathfrak{g}$ in each datasets' expression component $\mathbf{X}^1, \mathbf{X}^2$. Using HiRef output $T$, we also formed the *predicted* abundance vector $\hat{\mathbf{v}}^{\mathfrak{g}}$, which maps the raw counts from $\mathbf{v}^{1,\mathfrak{g}}$ to the spots in the second dataset through $T$.

Moreover, to compute cosine similarities between predicted and true expression abundances, (Clifton et al., 2023) employ a spatial binning on their output, using windows of $200\mu$m to tile each slice. The diameter of each slice is roughly $10,000\mu$m, and to make our output comparable, we used the spatial coordinates $\mathbf{S}'$ to bin and average the vectors $\mathbf{v}^{2,\mathfrak{g}}$ and $\hat{\mathbf{v}}$ locally. We used a total of 5625 bins, corresponding to a 15-to-1 mapping from spots to bins. Averaging the abundance of gene $\mathfrak{g}$ in each bin, we obtain spatially smoothed versions of $\mathbf{v}^{2,\mathfrak{g}}$ and $\hat{\mathbf{v}}$, as in (Clifton et al., 2023). Denote these smoothed vectors by $\mathbf{w}^{2,\mathfrak{g}}$ and $\hat{\mathbf{w}}$. For each gene $\mathfrak{g}$ among { *Slc17a7, Grm4, Olig1, Gad1, Peg10* }, we computed the cosine similarity between $\mathbf{w}^{2,\mathfrak{g}}$ and $\hat{\mathbf{w}}$, listing our results in Table S7. In the same table, we list scores obtained by the low-rank methods FRLC (Halmos et al., 2024) and LOT (Scetbon et al., 2021) for comparison. While HiRef is restricted to running on datasets of the same size, LOT and FRLC have no such restriction, and can run on the pair of MERFISH slices without any subsampling. To address this, in each case of LOT and FRLC, we give results from the methods run on the datasets with *and* without subsampling, reporting the highest scores for each method in main. In particular, we compared the cosine similarities for the original and sub-sampled dataset on a downstream task, as the primal OT cost is no longer directly comparable. Without the sub-sampling, the cosine score is only slightly higher than with: (0.3390, 0.2712, 0.3186, 0.1666, 0.1080) vs (0.3241, 0.2279, 0.3029, 0.1653, 0.0719). These scores remain significantly lower than those of hierarchical refinement on the sub-sampled data: (0.8098, 0.7959, 0.7526, 0.4932, 0.6015).

For the FRLC algorithm, we set $\alpha = 0$, $\gamma = 200$, $\tau_{\text{in}} = 500$, rank $r = 500$, using 20 outer iterations and 300 inner iterations. The runtime of FRLC was 1 minute 26 seconds on an A100 GPU. For the LOT algorithm, we were unable to pass a low-rank factorization of the distance matrix, so we had to use a smaller rank $r = 20$ in order to avoid exceeding GPU memory (the choice $r = 20$ led to memory usage of 30GB). We set $\epsilon = 0.01$ and otherwise used the default settings of the method. The

*Table S7.* Cosine Similarity Scores for Expression Transfer & Spatial Transport Cost

| Method | Slc17a7 | Grm4 | Olig1 | Gad1 | Peg10 | Transport Cost |
|---|---|---|---|---|---|---|
| `HiRef` (this work) | **0.8098** | **0.7959** | **0.7526** | **0.4932** | **0.6015** | **330.3301** |
| FRLC (Halmos et al., 2024) | 0.2180 | 0.2124 | 0.1929 | 0.0963 | 0.0991 | 415.0683 |
| FRLC, no subsampling | 0.2373 | 0.1896 | 0.1579 | 0.0644 | 0.1550 | 634.4158 |
| LOT (Scetbon et al., 2021) | 0.3241 | 0.2279 | 0.3029 | 0.1653 | 0.0719 | 3722.3171 |
| LOT, no subsampling | 0.3390 | 0.2712 | 0.3186 | 0.1666 | 0.1080 | 3722.1360 |
| MOP (Gerber & Maggioni, 2017) | 0.5211 | 0.4714 | 0.5972 | 0.3571 | 0.2719 | 2479.6117 |
| Mini-batch (128) | 0.6693 | 0.6637 | 0.6442 | 0.4150 | 0.4932 | 653.0491 |
| Mini-batch (512) | 0.7089 | 0.7383 | 0.6771 | 0.4562 | 0.5383 | 438.1703 |
| Mini-batch (1,024) | 0.7256 | 0.7621 | 0.6918 | 0.4733 | 0.5557 | 384.2498 |
| Mini-batch (2,048) | 0.7434 | 0.7822 | 0.7056 | 0.4912 | 0.5683 | 349.2964 |

*Table S8.* Cost Values $\langle \mathbf{C}, \mathbf{P} \rangle_F$ for ImageNet (Deng et al., 2009; Russakovsky et al., 2015) Alignment Task.

| Method | HiRef | Sinkhorn | MB 128 | MB 256 | MB 512 | MB 1024 | FRLC | LOT | ProgOT |
|---|---|---|---|---|---|---|---|---|---|
| **OT Cost** | **18.97** | N/A | 21.89 | 21.11 | 20.34 | 19.58 | 24.12 | N/A | N/A |

total runtime was 36 minutes 8 seconds on an A100 GPU. To form a spot-to-spot mapping from each transport plan output by FRLC and LOT, we mapped the spot with index $i$ in the first slice to the index argmax of the $i$-th row of the transport plan. Note that we ran LOT using the squared Euclidean cost as default, as passing `cost_fn=costs.Euclidean()` as an argument to `ott-jax`'s `PointCloud` raised an error. The discrepancy in transport cost between the two low rank methods reported in Table S7 is explained by (i) needing to use squared-Euclidean cost in the case of LOT, and (ii) using a rank-20 plan of LOT versus the rank-500 plan of FRLC. We applied the exact same spatial averaging to the outputs of all methods. We plot the ground-truth and `HiRef`-predicted abundances in Figure S1.

### D.4. Alignment of ImageNet Embeddings

To demonstrate the scalability of `HiRef` to massive and high-dimensional datasets, we perform an alignment unprecedented for OT solvers: aligning 1.281 million images from the ImageNet ILSVRC dataset (Russakovsky et al., 2015; Deng et al., 2009). A negligible amount of sub-sampling, 167 points out of 1281167, was applied so that $n$ divided into two integers $n/2 = 640500$ of which neither is prime. From this, `rank_annealing.optimal_rank_schedule(n, hierarchy_depth, max_Q, max_rank)` was called to generate the depth 3 rank-annealing schedule of $(r_1, r_2, r_3) = (7, 50, 1830)$ for `HiRef`. We used the ResNet50 architecture (He et al., 2016) available at https://download.pytorch.org/models/resnet50-0676ba61.pth to generate embeddings of each image of dimension $d = 2048$. We then took a 50:50 split of the dataset as the two image datasets $\mathsf{X}, \mathsf{Y}$ to be aligned, where we used a random permutation of the indices of the dataset using `torch.randperm` so that the splits approximately represent the same distribution over images. We then aligned these image datasets using `HiRef` FRLC, and mini-batch OT. For $\mathbf{z}_i, \mathbf{z}_j \in \mathbb{R}^{2048}$ we used the standard Euclidean cost defined by

$$\mathbf{C}_{ij} = \|\mathbf{z}_i - \mathbf{z}_j\|_2$$

We use the sample-linear algorithm (Indyk et al., 2019) to factorize $\mathbf{C}$ into low-rank factors of dimensions $(d_1, d_2, d_3) = (r_1, r_2, r_3) = (7, 50, 1830)$ paralleling the rank-schedule. The final cost values for each are shown in Table S8.

## E. Additional Information

There are a number of additional practical details regarding Algorithm 1 in its actual implementation. In particular, to achieve linear scaling, one must also have sample-linear approximation of the distance matrix $\mathbf{C}$. We use the algorithm of (Indyk et al., 2019) to accomplish this, as discussed in Section E.1. In addition, one requires parallel sequence of ranks for the distance matrices used at each step, $(d_1, \cdots, d_\kappa)$. As a default, we set $(d_1, \cdots, d_\kappa) = (r_1, \cdots, r_\kappa)$ so that the ranks of the distance matrices parallel those of the coupling matrices. Moreover, `HiRef` has the capacity to be heavily parallelized: since Algorithm 1 breaks each instance into independent partitions, one may also parallelize the low-rank sub-problems of

Table S9. Hyperparameters for ImageNet Experiment

| Parameter Name | Variable | Value |
|---|---|---|
| Rank-Annealing Schedule | $(r_1, \ldots, r_\kappa)$ | [7, 50, 1830] |
| Hierarchy Depth | $\kappa$ | 3 |
| Maximal Base Rank | $Q$ | $2^{11}$ |
| Maximal Intermediate Rank | $C$ | 64 |

Algorithm 1 across compute nodes.

### E.1. Optimizing the Rank-Annealing Schedule

As discussed in Section 3.3, the large constants required by low-rank OT (LROT) in practice encourage factorizations which have *minimal* partial sums. In particular, one seeks a factorization which minimizes the number of times LROT is run as a sub-procedure. Suppose one defines the maximal admissible rank of the low-rank solutions to be $C \in \mathbb{Z}_+$, the hierarchy-depth to be $\kappa$, the number of data-points to be $n$, and the maximal-rank permissible for the base-case alignment to be $Q$. If $Q \neq 1$, then one may take $n \leftarrow n/Q$, $\kappa \leftarrow \kappa - 1$, to observe that the total number of runs required is $1 + r_1 + r_1 r_2 + \ldots + \prod_{i=1}^{\kappa} r_i$, where the ranks factor the sample-size as $\prod_{i=1}^{\kappa} r_i = n$. Thus, to optimize the number of LROT calls for a given hierarchy-depth $\kappa$, one can optimize for the rank-annealing schedule by minimizing the sum of partial products defined by $\min_{(r_i)_{i=1}^{\kappa}} \sum_{j=1}^{\kappa} \prod_{i=1}^{j} r_i$ subject to $\prod_{i=1}^{\kappa} r_i = n$, $r_i \in \mathbb{Z}_+$, $r_i \leq C$. Observing that this equals $\min_{(r_i)_{i=1}^{\kappa}} r_1 + r_1 \sum_{j=2}^{\kappa} \prod_{i=2}^{j} r_i$ implies a standard dynamic-programming approach and store a table of factors up to $C$ to optimize this in $O(C\kappa n)$ time for $C, \kappa$ generally pre-fixed constants.

**Low-rank distance matrix C.**  A key work (Indyk et al., 2019) showed that one may approximately factor a distance matrix $\mathbf{C}$ with linear complexity in the number of points $n$ (Algorithm E.1). For certain costs, e.g. squared Euclidean, this factorization can be given for free (Scetbon et al., 2021). We rely on both of these for low-rank factorizations of the distance matrix, so that both the space of the coupling and pairwise distance matrix scale linearly.

---

**Algorithm 3**  Low-Rank approximation for distance matrix $\mathbf{C}$

---

Input point sets $\{\mathbf{x}_i\}_{i=1}^{n}$, $\{\mathbf{y}_j\}_{j=1}^{M}$ in metric space $\mathcal{X}$ and metric $d$
Pick indices $i^* \in [n], j^* \in [m]$ uniformly at random
**for** $i = 1$ to $n$ **do**
    Update sample probability $p_i = d(\mathbf{x}_i, \mathbf{y}_{j^*})^2 + d(\mathbf{x}_{i^*}, \mathbf{y}_{j^*})^2 + \frac{1}{m} \sum_{j=1}^{m} d(\mathbf{x}_{i^*}, \mathbf{y}_j)^2$
**end for**
Sample $O(r/\varepsilon)$ rows $\mathbf{C}_{i,.} \sim Categorical\left(\frac{p_i}{\sum_i p_i}\right)$
Compute $\mathbf{U}$ using (Frieze et al., 2004)
Compute $\mathbf{V}$ using (Chen & Price, 2017)
return $\mathbf{V}, \mathbf{U}$

---

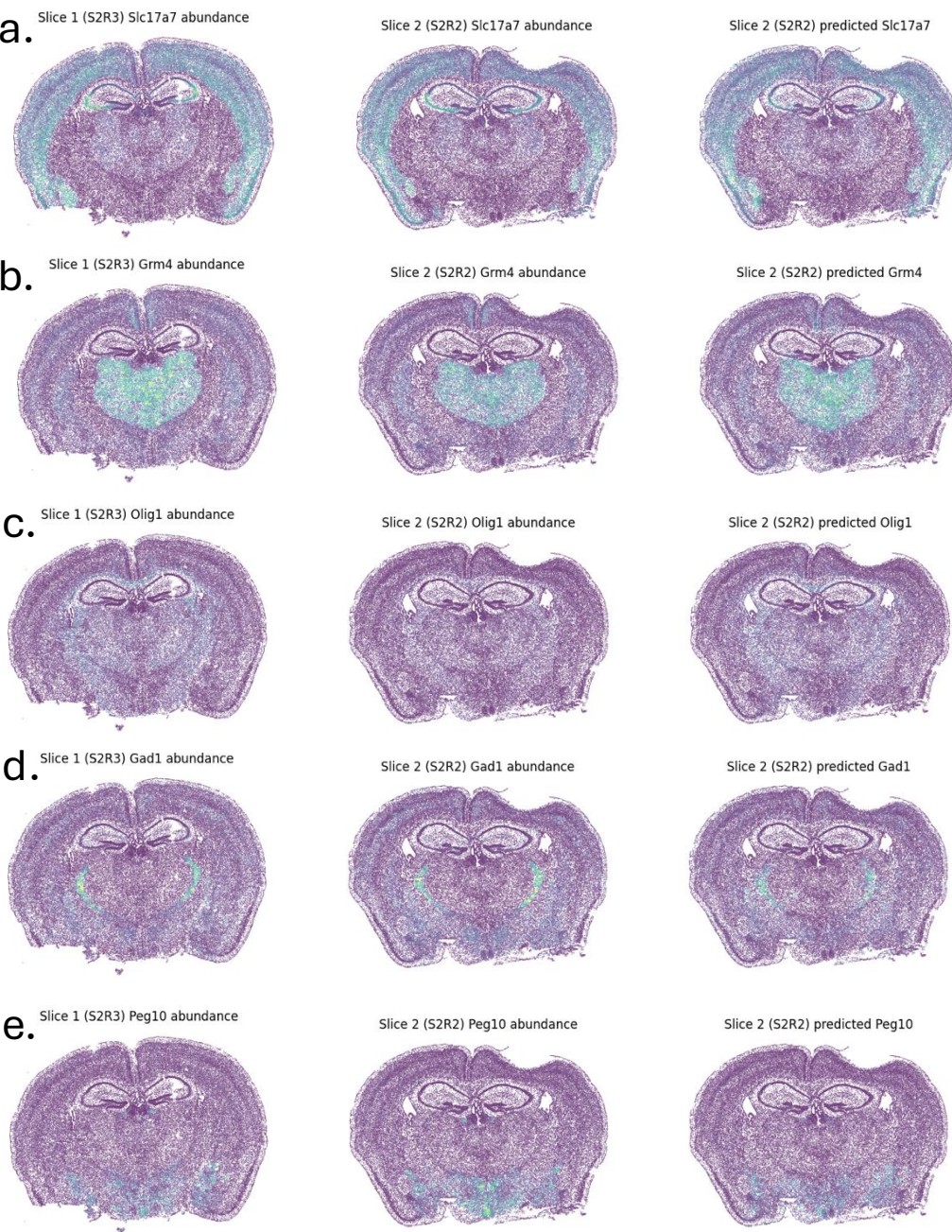

*Figure S1.* Abundance of 5 genes (**a.** *Slc17a7*, **b.** *Grm4*, **c.** *Olig1*, **d.** *Gad1*, **e.** *Peg10*) in Allen Brain Atlas MERFISH dataset (Clifton et al., 2023). From left to right are plotted (1) abundance in the first dataset, (2) abundance in the second dataset, and (3) predicted abundance via transfer of the abundances in the first dataset under the mapping of `HiRef`.

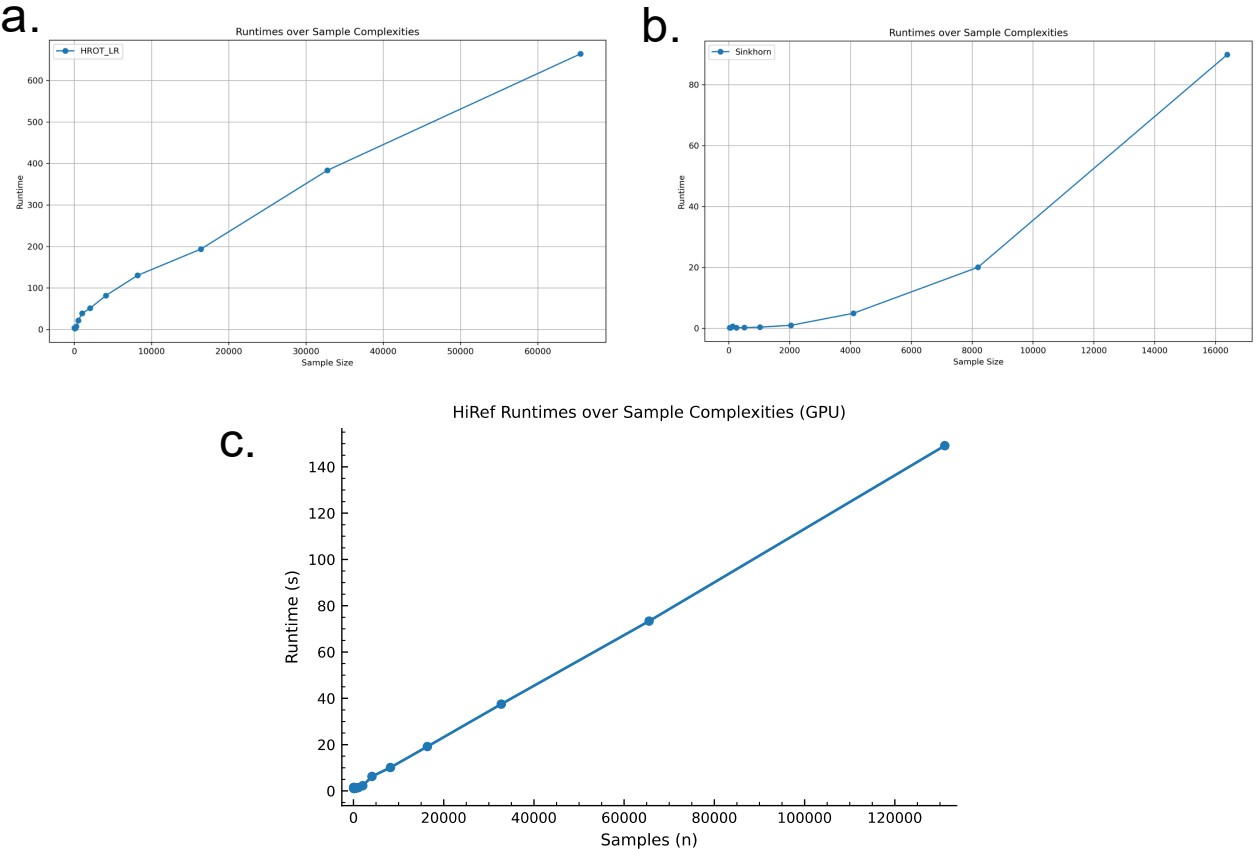

*Figure S2.* Runtime scaling across sample-complexities $n$ of **a.** Hierarchical Refinement (`HiRef`) and **b.** Sinkhorn for Euclidean cost, $\|\cdot\|_2$ (single CPU core). Hierarchical Refinement exhibits linear scaling for increasing $n$, whereas Sinkhorn exhibits quadratic scaling and is unable to run beyond 16k points. **c.** Runtime scaling of `HiRef` (GPU).

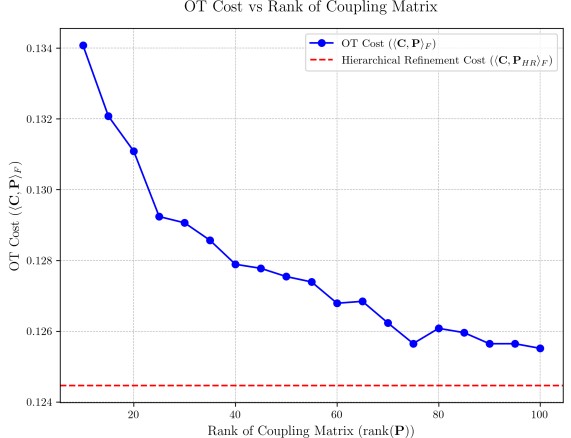

*Figure S3.* `HiRef` cost and the cost of the low-rank OT solution of FRLC (Halmos et al., 2024) across the coupling rank $r \in [5, 100]$.

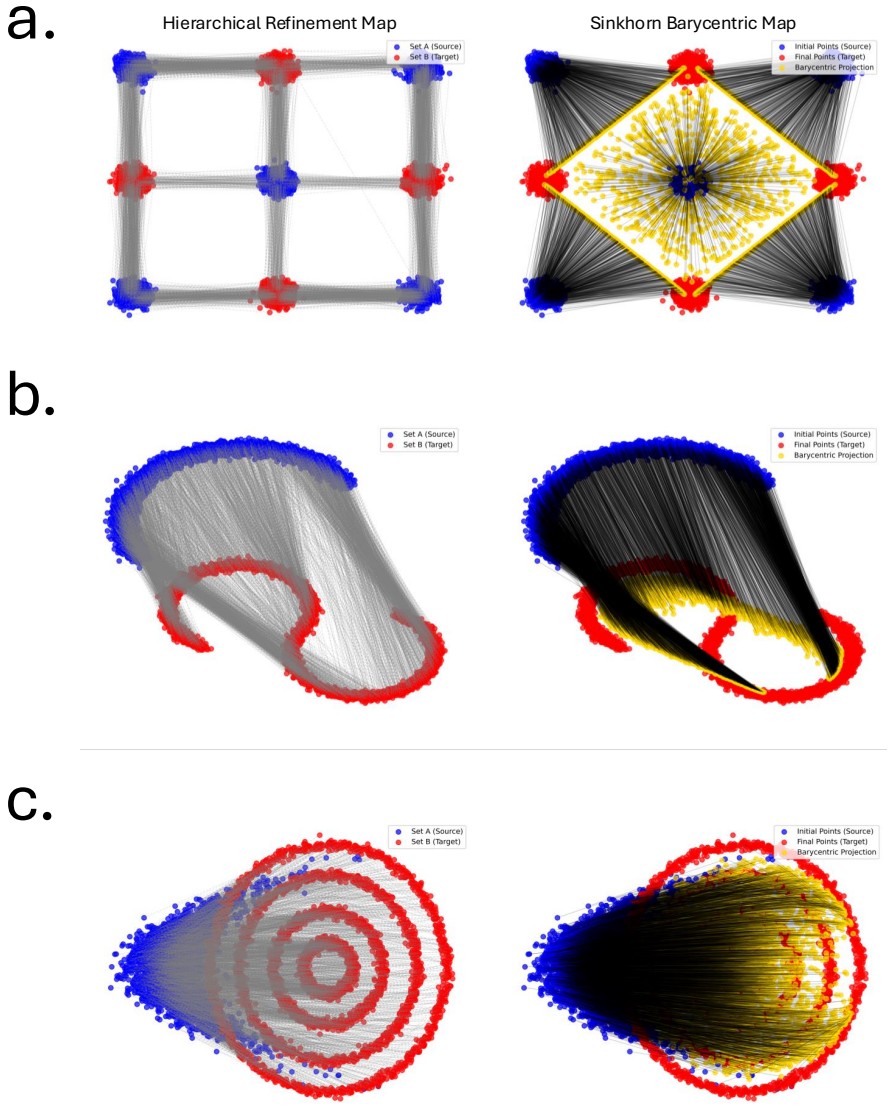

*Figure S4.* Comparison of optimal transport maps under (1) the `HiRef` alignment, and (2) the Sinkhorn (Cuturi, 2013) barycentric projection. **a.** The checkerboard dataset of (Makkuva et al., 2020), **b.** the Half-moon and S-curve dataset of (Buzun et al., 2024), and **c.** the MAF-Moons Rings dataset of (Buzun et al., 2024).

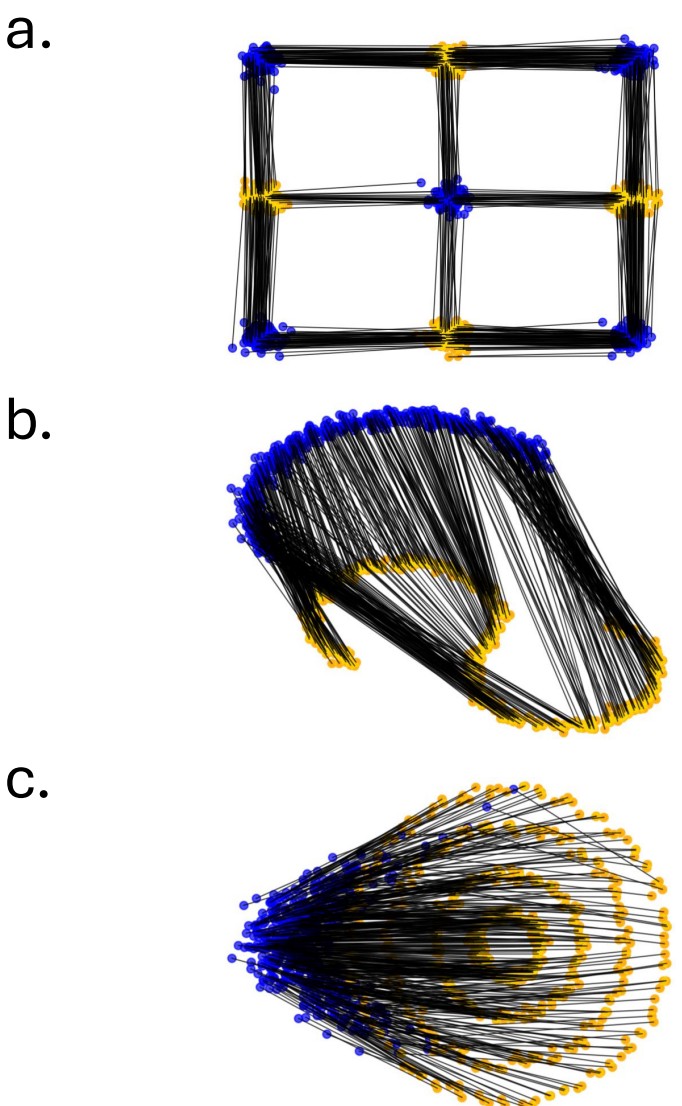

*Figure S5.* Alignments of the synthetic datasets of (Makkuva et al., 2020; Buzun et al., 2024) using the optimal dual revised simplex (Huangfu & Hall, 2018) algorithm for small instances (512 points). **a.** The checkerboard dataset of (Makkuva et al., 2020), **b.** the Half-moon and S-curve dataset of (Buzun et al., 2024), and **c.** the MAF-Moons Rings dataset of (Buzun et al., 2024).

