# OpenReview forum: "Hierarchical Refinement: Optimal Transport to Infinity and Beyond"
_ICML.cc/2025/Conference — ICML 2025 oral_

### Official Review · Reviewer_9vCU · 2025-03-10

**Overall Recommendation:** 4

**Summary:**

This paper proposes a new large-scale hierarchical optimal transport algorithm, between distributions with the same number of samples. The authors algorithm is based on multi-scale partitions of the source and target datasets, as well as recent development in low-rank optimal transport. Through a series of experiments on synthetic and real-world datasets, the authors show that their approach 1) has better alignment performance than previously proposed methods and 2) can scale to larger datasets.

**Claims And Evidence:**

There are 2 main claims in this paper.

1) The proposed strategy successfully aligns source and target distributions,
2) The algorithm can scale to larger datasets

Both of them are supported in the experiments. Furtheremore, there are some theoretical results supporting 1).

**Essential References Not Discussed:**

As I mentioned in the previous point, I think the paper could benefit from a broader discussion with hierarchical optimal transport, especially these references,

(Chen, Georgiu and Tannenbaum, 2017) Chen, Yongxin, Tryphon T. Georgiou, and Allen Tannenbaum. "Optimal transport for Gaussian mixture models." IEEE Access 7 (2018): 6269-6278.

(Yurochkin et al, 2018) Yurochkin, Mikhail, et al. "Hierarchical optimal transport for document representation." Advances in neural information processing systems 32 (2019).

(Delon and Desolneux, 2019) Delon, Julie, and Agnes Desolneux. "A Wasserstein-type distance in the space of Gaussian mixture models." SIAM Journal on Imaging Sciences 13.2 (2020): 936-970.

(El Hamri, Bennani and Falih, 2021) El Hamri, Mourad, Younes Bennani, and Issam Falih. "Hierarchical optimal transport for unsupervised domain adaptation." Machine Learning 111.11 (2022): 4159-4182.

Furthermore, while the authors do mention mini-batch OT, the authors could have cited a few papers on the problem, such as,

(Nguyen et al., 2022) Khai Nguyen, Dang Nguyen, The-Anh Vu-Le, Tung Pham, Nhat Ho Proceedings of the 39th International Conference on Machine Learning, PMLR 162:16656-16690, 2022.

(Fatras et al., 2022) Kilian Fatras, Thibault Sejourne, Rémi Flamary, Nicolas Courty Proceedings of the 38th International Conference on Machine Learning, PMLR 139:3186-3197, 2021.

(Fatras et al., 2020) Fatras, Kilian, et al. "Learning with minibatch Wasserstein: asymptotic and gradient properties." International Conference on Artificial Intelligence and Statistics. PMLR, 2020.

including hierarchical approaches

(Nguyen et al., 2022) Khai Nguyen, Dang Nguyen, Quoc Dinh Nguyen, Tung Pham, Hung Bui, Dinh Phung, Trung Le, Nhat Ho Proceedings of the 39th International Conference on Machine Learning, PMLR 162:16622-16655, 2022.

**Experimental Designs Or Analyses:**

The experiments are valid and well designed. As I mentioned previously, they are comprehensive.

**Methods And Evaluation Criteria:**

The experiments are well designed and use relevant benchmarks. The authors could have considered other problems where OT has contributed to, such as generative modeling or domain adaptation, but overall the experiments are comprehensive.

**Other Comments Or Suggestions:**

In Figure 2, the x-axis should be in log-scale

**Other Strengths And Weaknesses:**

Here, I make a summary of my review.

**Strenghts**

1. The paper provides a practical and scalable algorithm for large-scale optimal transport
2. The idea of using multi-level partitions is quite interesting
3. The discussion at the end of the paper is quite comprehensive and deals with natural questions that arise from the reading of the paper.

**Weaknesses**

1. The authors could have considered minibatch-OT in their experimental comparisons. There could be a deeper discussion of minibatch OT in the main paper

----

As a result, I am leaning towards acceptance, as I think the paper is good.

---

__Post Rebuttal:__ The authors addressed my concerns in their rebuttal. As a result, I raise my score from __3. Weak Accept__ to __4. Accept__

**Questions For Authors:**

Here are a few questions for the authors,

- Do the linear space complexity stems from the assumption that $X$ and $Y$ have the same number of elements? For instance, in principle, if $X$ and $Y$ have the same number of elements and have uniform weights, one can store $\{x_{i}, T(x_{i})\}_{i=1}^{n}$ with classic OT.
- What would be the challenges of adapting their method to handle distributions with unequal number of samples?

**Relation To Broader Scientific Literature:**

This paper fits into the broader litreature of Hierarchical OT (Chen, Georgiu and Tannenbaum, 2017; Yurochkin et al, 2018; Delon and Desolneux, 2019; El Hamri, Bennani and Falih, 2021), which consider transportation plans at different levels of representations of probability measures through clustering. Arguably, the paper pushes this idea further, as it consider partitions of increasing depth of the source and target datasets.

**Theoretical Claims:**

There are 3 main theoretical results (Propositions 3.1, 3.2 and 3.3). I could not assess the validity of these proofs. I have some questions down below my review.

---

> ### Author Rebuttal · Authors · 2025-04-01
>
> We thank reviewer _9vCU_ for their feedback and careful reading.
>
> > As I mentioned ... authors could have cited a few papers ...
>
> Thank you for these suggestions, we will include the following sentences in our Background:
> _"Hierarchical OT (Schmitzer and Schnorr '13) is a variant of OT modeling data and transport across multiple scales, using Wasserstein distances as coarse-scale ground costs. It has been applied to document representation (Yurochkin et al. '19), domain adaptation (El Hamri et al. '22), sliced Wasserstein distances (Bonneel et al. '15; Nguyen et al. '22b) and to give a discrete formulation of transport between Gaussian mixture models (Chen et al. '18; Delon and Desolneux '20) . These works build interpretable, coarse-grained structure into a single transport plan, rather than solving for a sequence of plans at progressively finer scales as in the present work."_
>
> In addition, we have added experimental validation against mini-batch OT. We will add the following sentences to our Introduction:
> _"Mini-batch OT (Genevay et al. '18) improves scalability, but incurs significant biases (Sommerfeld et al. '19; Korotin et al. '21; Fatras et al. '21a) as each mini-batch alignment is often a poor representation of the global one. Several works have investigated the theoretical properties of mini-batch estimators of the plan (Fatras et al. '20b; '21c), while others have attempted to mitigate bias using partial or unbalanced OT (Nguyen et al. '22a; Fatras et al. '21b) by allowing for mass variation between the mini-batches."_
>
> >  ..deeper discussion of minibatch OT..
>
> Thank you for pointing this out. We re-ran the large-scale real data experiments with mini-batch OT without-replacement, which is more conventional as noted in (Fatras et al. '21c), and will include these values in the updated manuscript. See the tables for the mouse embryo spatial transcriptomics, ImageNet below and MERFISH experiments above in response to _nCzR_, including mini-batch couplings. We notate with (MB $B$) for $B$ the batch size in each case. One value in the HR-OT table (Cost 14.35 for 12.5-13.5) has been changed, and reflects a slightly higher setting of $r _ {max} = 128$ in the rank annealer. We also will add more discussion to our paper on practical considerations for mini-batch OT, as well as existing theoretical results on convergence and complexity (Fatras et al. '20a; '21a; '21c) relative to full-rank OT.
>
> **Table: Cost Values for Embryo**
>
> |Method|9.5-10.5|10.5-11.5|11.5-12.5|12.5-13.5|13.5-14.5|14.5-15.5|15.5-16.5|
> |-|-|-|-|-|-|-|-|
> |HR-OT|**21.81**|**14.81**|**16.14**|**14.35**|**13.78**|**14.29**|**12.79**|
> |MB 128|22.44|15.35|16.69|14.86|14.14|14.75|13.32|
> |MB 512|22.15|15.05|16.33|14.54|13.92|14.50|13.01|
> |MB 1024|22.05|15.02|16.24|14.45|13.86|14.43|12.91|
> |MB 2048|21.98|14.98|16.18|14.39|13.81|14.39|12.85|
>
> **Table: Cost Values for ImageNet**
>
> |Method|HR-OT|MB 128|MB 512|MB 1024|
> |-|-|-|-|-|
> | |**18.97**|21.89|20.34|19.58|
>
>
> We will also log-transform Fig.2's $x$-axis for the next version!
>
> > Do the linear space complexity..?
>
> Great question. While you are correct that bijections can be stored with linear space, our linear space complexity does not rely on $X$ and $Y$ having the same number of elements. For classical approaches tackling OT as an assignment problem, while the final coupling returned is linear in space, the space complexity of the algorithm is quadratic. In comparison, HR-OT returns a solution with linear space and has linear space complexity. In addition, the runtime complexity is log-linear for squared-Euclidean cost $\lVert \cdot \lVert_{2}^{2}$ and remains log-linear for sample-linear approximations of the distance matrix (Indyk et al. '19). We refer to the bullet below on your question concerning whether hierarchical refinement could retain its properties with an unequal number of points.
>
> > What would be the challenges of adapting their method to handle distributions with unequal number of samples?
>
> There are two challenges to extend hierarchical refinement to datasets with $n$ source points and $m$ target points.
>
> 1. If $n > m$, the challenge is to extend Proposition 3.1 from 1-1 Monge maps (the assignment problem) to many-to-one Monge maps (i.e. with smaller target dataset). Specifically, one needs to account for the possibility that each target index $j \in [m]$ is co-clustered with a _set_ $ S _ {j} \subset [n]$ of indices indexing the preimage of the Monge map $T^{-1}( \mathbf{x} _ {j} )$.
>
> 2. If $m>n$, no Monge map exists from the source to target dataset. While Sinkhorn approaches the Kantorovich problem, hierarchical refinement approaches the Monge formulation of optimal transport. Thus, it inherits any limitations of the Monge framework, including its asymmetry and inability to account for mass-splitting. However, assuming the extension of the proposition discussed in point (1.) holds, one can reverse the role of the two datasets and infer the Monge map from target to source.

---

> > ### Comment · Reviewer_9vCU · 2025-04-03
> >
> > Thank you for your rebuttal.
> >
> > I consider that your comments answer my questions, and I will raise my score accordingly, from __3. Weak Accept__ to __4. Accept__. Congratulations on your work.
> >
> > In the meantime, I want to raise some discussion for future works on extending the authors' work to $n \neq m$. While the authors are correct on their remark about the challenges of adapting their strategy for $n \neq m$, they could find an approximation for the actual monge map using the barycentric projection,
> >
> > $$T\_{\gamma}(x\_{i}) = \text{argmin}\_{x \in \mathbb{R}^{d}}\sum\_{j=1}^{n}c(x,y\_{j})\gamma\_{ij},$$
> >
> > which, for the squared Euclidean cost, has closed-form solution. This mapping has been widely used, for instance, in domain adaptation. The authors can consult [R1] and the references therein for further information about how this mapping approximates the Monge map.
> >
> > Note that this comment is surely beyond the scope of the current paper, and have nothing to do with the current evaluation.
> >
> > [R1] Deb, Nabarun, Promit Ghosal, and Bodhisattva Sen. "Rates of estimation of optimal transport maps using plug-in estimators via barycentric projections." Advances in Neural Information Processing Systems 34 (2021): 29736-29753.

---

### Official Review · Reviewer_j2d6 · 2025-03-11

**Overall Recommendation:** 4

**Summary:**

This paper proposes a hierarchical framework to obtain Kantarovich plans in Optimal Transport. The authors conceptually build on many prior works of low-rank OT (notably Scetbon 2021, Halmos 2024) and propose a rank annealing schedule to obtain the full rank Kantarovich plan using many low-rank solutions of sequentially partitioned sets. The main premise is founded on a theoretical result that when a Monge map exists between the datasets, it co-clusters the optimal low-rank solution (albeit with minor differences like uniform assumption in Eq 6). Altogether the proposed framework is claimed to be linear in space complexity and log-linear to quadratic in time complexity. The authors contrast this with well-known OT solvers mainly Sinkhorn (which has quadratic space and quadratic time) and ProgOT.

Experiments are reported for (1.) Synthetic Datasets (Checkerboard, MAFMoons and Rings, and Half-Moon and S-Curve etc) (2.) Large-scale Matching Problems and Transcriptomics (3.) MERFISHBrainAtlasAlignment and (4.) ImageNet Alignment. The authors compare with Sinkhorn (Cuturi, 2013), ProgOT (Kassraieetal.,2024), as well as low rank OT solvers like (Scetbon et al.,2021) and FRLC(Halmos et al.,2024). The main message from the experiments is to highlight that the proposed approach scales better (in space complexity) than existing full-rank solvers and produces a smaller kantarovich cost, establishing a good operating point between accuracy and complexity.

**Claims And Evidence:**

The paper is written well. The main contributions are highlighted nicely and the experiments reflect the nuanced benefits of the proposed approach. The main method is convincing - and I found it nice that the algorithm builds on Proposition 3.1

**Essential References Not Discussed:**

- I do feel the lack of *any* comparison to Gerber&Maggioni, 2017 to be underwhelming. Even though I agree it is not apples to apples comparison, this is a very strong and similar prior work, and perhaps an experiment highlighting the difference in performance for a reasonable choice of initial partition would be very enlightening

**Experimental Designs Or Analyses:**

Yes. All of the 4 experiments reported in the summary.

**Methods And Evaluation Criteria:**

Yes. The experiments are diverse and fairly comprehensive

**Other Comments Or Suggestions:**

- Eq 1 incomplete (missing b)
- Line 119, define  \delta_r

**Other Strengths And Weaknesses:**

Overall, barring some important clarifications (e.g., using the correct sinkhorn implementation), I am inclined positively. The paper has been compiled well, has an interesting message, and has comprehensive experiments.

**Questions For Authors:**

- Are the pointclouds in Figure 3 for comparing the 3 methods the *same*?
- Is the Sinkhorn used in the experiments (ott-jax) with the \eps annealer?
- This citation is missing: Cuturi, M., Meng-Papaxanthos, L., Tian, Y., Bunne, C., Davis, G., & Teboul, O. (2022). Optimal transport tools (ott): A jax toolbox for all things wasserstein. arXiv preprint arXiv:2201.12324.

**Relation To Broader Scientific Literature:**

- At a high level, the main message of the paper is to propose the use of low rank OT solvers to sequentially obtain a full rank solution. This submission does NOT propose a new low rank OT solver. If we credit the authors for focusing on this novelty - however, I find that the authors do make specific choices (1.) Use of Halmos et al in Algorithm 1 (2.) the specific Select subroutine in Algorithm 1. While the choices are not unreasonable, I do find the whole setup arguably lacking generality. For e.g., I could not place any experiment that validates replacing with Scetbon 2021 et al, etc. I feel it is most impactful when the novel idea is shown to work for different LROT solvers, which I am not yet convinced.

**Theoretical Claims:**

Yes. Propositions 3.1 3.2 and 3.3.

---

> ### Author Rebuttal · Authors · 2025-03-31
>
> We thank reviewer _j2d6_ for their feedback and careful reading.
>
> > At a high level, the main message of the paper is to propose the use of low rank OT solvers to sequentially obtain a full rank solution. This submission does NOT propose a new low rank OT solver. If we credit the authors for focusing on this novelty - however, I find that the authors do make specific choices (1.) Use of Halmos et al in Algorithm 1 (2.) the specific Select subroutine in Algorithm 1. While the choices are not unreasonable, I do find the whole setup arguably lacking generality. For e.g., I could not place any experiment that validates replacing with Scetbon 2021 et al, etc. I feel it is most impactful when the novel idea is shown to work for different LROT solvers, which I am not yet convinced.
>
> We agree on the benefit of demonstrating with other LROT solvers.  We did not have time to implement this in the short response window, but aim to offer this option in the code and demonstrate results with the Scetbon et al. 2021 solver. Theoretically, our framework will work for the Scetbon et al. solver because our Proposition 3.1 is tailored to the low-rank factorization introduced by (Scetbon et al. '21). Proposition 3.1 hinges on Lemma B1, which shows _optimal_ factors $(\mathbf{Q}, \mathbf{R})$ for the LOT problem (Scetbon et al. '21) are vertices on the transport polytope and thus have $\leq n+r-1=n+1$ non-zero entries for $r=2$. Thus for $n = 2^t$ the solutions correspond to partitions in Proposition 3.1. The key point is these arguments are independent of the solver, and apply to any optimal solution.
>
> Regarding (2), our choice of the argmax as the Select sub-routine is motivated by our Proposition 3.1, and is correct if $(\mathbf{Q}, \mathbf{R})$ are optimal, but other choices may be appropriate if the solutions are entropic and do not define exact partitions.
>
> > I do feel the lack of any comparison to Gerber & Maggioni, 2017 to be underwhelming. Even though I agree it is not apples to apples comparison, this is a very strong and similar prior work, and perhaps an experiment highlighting the difference in performance for a reasonable choice of initial partition would be very enlightening.
>
> We agree, Gerber & Maggioni is a seminal work in multi-scale OT, and we have added comparisons with their method MOP for our 2-dimensional datasets where MOP uses the GMRA (Geometric Multi-Resolution Analysis) R package to generate multiscale partitions. In particular, we benchmarked against MOP for our three synthetic datasets and our MERFISH expression transfer task. MOP's performance for MERFISH is given in the table above in response to reviewer _nCzR_. MOP's performance on synthetic data is given in the table below:
>
> **Table: Comparison of Coupling-Based OT Methods on Primal Cost $\langle \mathbf{C}, \mathbf{P} \rangle_F$ (Wasserstein-2) on 512 point small instance**
>
> | Method | Checkerboard | MAF Moons & Rings | Half Moon & S-Curve |
> |-|--|--|--|
> | MOP (Gerber et al. '17)  | 0.393 | 0.276 | 0.401   |
> | Sinkhorn (`ott-jax`)   | 0.136   | 0.221 | 0.338  |
> | ProgOT (Kassraie et al. '24)  | 0.136  | 0.216 | 0.334   |
> | HR-OT  | 0.129        | 0.216  | 0.334               |
> | Dual Revised Simplex Solver   | **0.127** | **0.214** | **0.332**   |
>
> > Eq 1 incomplete (missing b); Line 119, define $\delta_r$
>
> Thank you for catching these mistakes. We will fix them in the final version.
>
> > Are the pointclouds in Figure 3 for comparing the 3 methods the same?
>
> The left two pointclouds for  hierarchical refinement and Sinkhorn are the same, since these methods are able to run on 4096 points. The rightmost figure is for an optimal LP-solver, which scaled only to 512 points and thus is plotted with an identically distributed but smaller dataset. We will add this distinction to the legend.
>
> > Is the Sinkhorn used in the experiments (ott-jax) with the $\epsilon$ annealer?
>
> Regarding your question about the Sinkhorn implementation: we use `ott-jax` as the Sinkhorn solver, but do not use the $\epsilon$-annealer. We rely on the default implementation of Sinkhorn, which uses a fixed value of $\epsilon = 0.05$. This default value produces relatively sparse solutions. We will note this detail in the experimental section.
>
> > This citation is missing: Cuturi, M., Meng-Papaxanthos, L., Tian, Y., Bunne, C., Davis, G., \& Teboul, O. (2022). Optimal transport tools (ott): A jax toolbox for all things wasserstein. arXiv preprint arXiv:2201.12324.
>
> Thank you for catching this, we will add this citation after our references of `ott-jax`.

---

### Official Review · Reviewer_nCzR · 2025-03-14

**Overall Recommendation:** 3

**Summary:**

This paper focuses on the solving of Optimal Transformer (OT) problems.  To this end, it derives an algorithm, HR-OT that leverages the invariant under Monge map and dynamically constructs a multi-scale partition of each dataset using low-rank OT subproblems. By doing that, it could use linear space and achieve runtime ranging from log-linear to quadratic. Experiments have been conducted on several datasets, even including large scale data with over a million points. The results demonstrate its advantages over the original Sinkhorn.

**Claims And Evidence:**

Seems yes.

**Essential References Not Discussed:**

I have no suggestion about the reference list.

**Experimental Designs Or Analyses:**

Yes. The experimental designs are acceptable. However, it is better to include large scale real data, e.g., 3DMatch (https://3dmatch.cs.princeton.edu/) or  KITTI (https://www.cvlibs.net/datasets/kitti/eval_odometry.php). Moreover, as Sinkhorn has been widely used in point cloud registration algorithms, e.g., CoFiNet (NeurIPS 2021) and GeoTransformer (CVPR 2022), I would like to see the promotion of those methods by  replacing the Sinkhorn part with the proposed method.

**Methods And Evaluation Criteria:**

Seems yes.

**Other Comments Or Suggestions:**

See above. I highly suggest the authors to add the metioned experiments.

**Other Strengths And Weaknesses:**

Strengths:
1. The writing is fluent and the organization is also good.
2. The proposed method could be considered novel.
3. The proposed method could be applied on data with over a million points, which is highly valuable, if no one could achieve that before.

**Questions For Authors:**

1. On the MERFISH experiments, the score of HR-OT results are much higher than that of ther methods. However the reason under this should be further analyzed.

2. It has been metioned that the numer of sample should be the same (|X| = |Y| = n) for the proposed method，and it does not hold, the data could be slightly modified. Could the authors give an example how this could be done? And under this case, what is the results compared with other methods?

**Relation To Broader Scientific Literature:**

As a promoted version of Sinkhorn, it could use linear space and achieve runtime ranging from log-linear to quadratic. Validation is also provided via theoretical proof as well as experiments.

**Theoretical Claims:**

Yes. Should be correct.

---

> ### Author Rebuttal · Authors · 2025-03-31
>
> We thank reviewer _nCzR_ for their feedback and careful reading.
>
> > Yes. The experimental designs are acceptable. However, it is better to include large scale real data, e.g., 3DMatch (https://3dmatch.cs.princeton.edu/}{https://3dmatch.cs.princeton.edu/) or KITTI (https://www.cvlibs.net/datasets/kitti/eval_odometry.php). Moreover, as Sinkhorn has been widely used in point cloud registration algorithms, e.g., CoFiNet (NeurIPS 2021) and GeoTransformer (CVPR 2022), I would like to see the promotion of those methods by replacing the Sinkhorn part with the proposed method.
>
> Thank you for your response. We will investigate using Hierarchical Refinement as a module in CoFiNet and GeoTransformer, but unfortunately have limited time in the review period to benchmark it. We will, however, make sure to cite both of these methods and highlight them as a major application area for hierarchical refinement in our discussion with the following sentence:
>
> * _"Optimal transport has also been successfully integrated into deep-learning frameworks for computer vision and point cloud registration, notably in methods such as CoFiNet (Yu et al. '21) and GeoTransformer (Qin et al. '22), suggesting that hierarchical refinement could help scale existing deep-learning methods based on OT."_
>
> On the scalability of hierarchical refinement on real datasets, we note that the size of the real datasets used was 85958 and 84172 for the two MERFISH datasets, 5913, 18408, 30124, 51365, 77369, 102519, 113350, and 121767 for the seven Stereo-Seq spatial transcriptomics datasets, and 1.281 million for the ImageNet ILSVRC dataset.
>
> > On the MERFISH experiments, the score of HR-OT results are much higher than that of ther methods. However the reason under this should be further analyzed.
>
> In the MERFISH experiments, as classical full-rank methods were unable to scale, we benchmarked against low-rank OT methods for a fixed rank. Low-rank OT is unable to compute one-to-one alignments in the case that the Monge map exists, and in this case an alignment of spatial coordinates lacks any low-rank cluster structure, which likely explains why full-rank performs significantly better. To address questions raised by other reviewers, we added a comparison to mini-batch OT on this task. Mini-batch OT is also a full-rank method, like hierarchical refinement, and we observe scores which are much closer to it. For example, for the largest batch-size (2048) we found mini-batch cosine scores across 5 genes of (0.7434, 0.7822, 0.7056, 0.4912, 0.5683), compared to the hierarchical refinement scores (0.8098, 0.7959, 0.7526, 0.4932, 0.6015). This implies the gap may largely explained by the difference in expressivity of full-rank ($r=84,172$) versus low-rank couplings ($r=20$ for LOT, $r=500$ for FRLC).
>
> > It has been mentioned that the number of samples ... what is the results compared with other methods?
>
> Thank you for your question. To your point, if the datasets are of slightly different sizes one could randomly subsample the number of points in the larger one to ensure an $n$ to $n$ alignment. In all comparisons, the datasets aligned are either $n \times n$ or sub-sampled to be $n \times n$ so that all methods (Sinkhorn, ProgOT, FRLC, LOT, HR-OT, mini-batch) are compared on the exact same point clouds. To gauge the effect of subsampling the MERFISH data, we ran LOT without sub-sampling to $n$ points and then LOT with sub-sampling, comparing the cosine similarities on a downstream task, as the primal OT cost is no longer directly comparable. Without the sub-sampling, the cosine score is only slightly higher than with: (0.3390, 0.2712, 0.3186, 0.1666, 0.1080) vs (0.3241, 0.2279, 0.3029, 0.1653, 0.0719). These scores remain significantly lower than those of hierarchical refinement on the sub-sampled data: (0.8098, 0.7959, 0.7526, 0.4932, 0.6015). While sub-sampling incurred little error on this comparison, generalizing hierarchical refinement to directly handle datasets of unequal sizes (i.e. without sub-sampling) is an important direction for future work.
>
> **Table: Cosine Similarity Scores for Expression Transfer**
>
> | Method  | *Slc17a7* | *Grm4*   | *Olig1*  | *Gad1*   | *Peg10*  |
> |-|-|-|-|-|-|
> | HR-OT   | **0.8098**| **0.7959**| **0.7526**| **0.4932**| **0.6015**|
> | FRLC (Halmos et al. '24)  | 0.2180    | 0.2124   | 0.1929   | 0.0963  | 0.0991   |
> | FRLC, no subsampling | 0.2373    | 0.1896   | 0.1579   | 0.0644 | 0.1550   |
> | LOT (Scetbon et al. '21)  | 0.3241    | 0.2279   | 0.3029   | 0.1653  | 0.0719   |
> | LOT, no subsampling  | 0.3390    | 0.2712   | 0.3186   | 0.1666 | 0.1080   |
> | MOP (Gerber et al. '17)  | 0.5211    | 0.4714 | 0.5972   | 0.3571 | 0.2719   |
> | Mini-batch (128)  | 0.6693 | 0.6637   | 0.6442 | 0.4150   | 0.4932 |
> | Mini-batch (512)  | 0.7089 | 0.7383   | 0.6771 | 0.4562   | 0.5383 |
> | Mini-batch (1,024)  | 0.7256 | 0.7621| 0.6918 | 0.4733   | 0.5557 |
> | Mini-batch (2,048)  | 0.7434 | 0.7822| 0.7056 | 0.4912   | 0.5683 |

---

### Official Review · Reviewer_dsLw · 2025-03-14

**Overall Recommendation:** 5

**Summary:**

This work concerns the use of hierarchical refined version of low-rank optimal transport (HR-LOT). In previous works, low-rank transportation plans have been explore to reduce (high) computational costs for solving the optimal transport problem, and hierarchically refined version of this problem was also studied. This work combines the two,

**Claims And Evidence:**

The authors claim that for optima transportation problems that do not necessarily possess the low-rank structure as assumed by LOT, HR-LOT enriches the space of transport maps while maintaining the linear runtime when low-rankness is present.

**Essential References Not Discussed:**

I'm not aware of paper the authors missed.

**Experimental Designs Or Analyses:**

The experiments are performed mostly on existing datasets, ranging from synthetic to realistic.

**Methods And Evaluation Criteria:**

The examples are illustrative and convincing.

**Other Comments Or Suggestions:**

- Hierarchical low-rankness of the transportation map seems to share common properties with so-called Hierarchical Matrices used to solve partial differential equations (most notably the Helmholtz equation). Are there some commonality in their approaches?

**Other Strengths And Weaknesses:**

Strengths

The paper is very well motivated and clearly written, and arguments are convincing. The experiments are thorough and the discussion is sufficiently detailed.

Weaknesses

The theoretical result Proposition 3.1 seems to have strong assumptions about the transportation plan, although a more general result might be simply very challenging to formulate and prove.

**Questions For Authors:**

I found the discussion regarding the rank-scheduling and annealing scheduling $\epsilon_n$ in Section 3.2 a bit hard to follow. Is there an alternative explanation for their relation?

**Relation To Broader Scientific Literature:**

The paper is an advance in LOT by adding the hierarchical partitioning. This in itself is novel, using algorithms both in hierarchical OT and LOT, and putting it together with a rank-annealing scheduling in Algorithm 1.

**Theoretical Claims:**

The Propositions 3.1-3 appear to be correct, although the assumptions in the statements are strong.

---

> ### Author Rebuttal · Authors · 2025-03-31
>
> We thank reviewer _dsLw_ for their feedback and careful reading.
>
> > The theoretical result Proposition 3.1 seems to have strong assumptions about the transportation plan, although a more general result might be simply very challenging to formulate and prove.
>
> Yes, two important generalizations of  Proposition 3.1 are: (1) extend to datasets of unequal sizes $n \neq m$;  (2) extend to general schedules of factors (as is done in practice in the algorithm) rather than just powers of 2. We briefly discuss (1) in response to reviewer 9vCU below. Regarding (2), we note requiring that $n$ be a power of two is without loss of generality in the Proposition. We will add the following statement to Section 3 to highlight this: _"We note that the assumption that the datasets are of size $2^k$ is without loss of generality. For a dataset of size $n$, let $q = \min$_ { _$2^{t} \mid t \in \mathbb{N}, 2^{t} > n $_ } _and add $q-n$ 'dummy' points at 'infinite' distance from $\mathsf{X}, \mathsf{Y}$, and mutual distance zero."_
>
> By construction, the optimal mapping for this augmented data is given as the product measure $\gamma = \gamma _ {1}^{ * } \otimes \gamma _ {2}^{ * }$, where $\gamma _ {1}^{ * }$ is the optimal mapping between the original points and $\gamma _ {2}^{ * }$ pairs the dummy points. Thus $\gamma_{1}^{ * }$ agrees with the original optimal coupling when restricted to the datasets $\mathsf{X}, \mathsf{Y}$, and one may thus invoke Proposition 3.1 directly for any $n$. The only computational cost is, in the worst case, one would extend the size of the dataset by a factor of 2.
>
> > Hierarchical low-rankness of the transportation map seems to share common properties with so-called Hierarchical Matrices used to solve partial differential equations (most notably the Helmholtz equation). Are there some commonality in their approaches?
>
> Yes! Thank you for highlighting this, as there appears to be an interesting parallel between the two. The coupling at each iteration $\mathbf{P} ^ {(t)}$ can be viewed as having a hierarchical block structure according to each partition. Similarly to how hierarchical matrices in PDE reduce the complexity of dense linear operators from $O(n^{2})$ to $O(n \log{n})$, hierarchical refinement uses a multiscale block structure to reduce the complexity of OT to $O(n)$ in space, and $O(n\log{n})$ in time. For iterations $t \in [0, \log{n} ]$ earlier iterations capture coarser structure, while later iterations capture fine-grained structure.
>
> A noteworthy difference is that the linear operators approximated by hierarchical matrices in PDE are inherently dense, while the optimal solution to OT is guaranteed to be sparse. In OT, dense matrices are prevalent because of the computational value of entropic (or rank) regularization as a means to explore the solution space by annealing from a dense initial condition to an optimal sparse solution.
>
> > I found the discussion regarding the rank-scheduling and annealing scheduling $\epsilon_n$ in Section 3.2 a bit hard to follow. Is there an alternative explanation for their relation?
>
> Thank you for this feedback. First, to better explain the connection between entropy- and rank-regularization, we refer to Chapter 5 in the thesis of Scetbon, based on the seminal work (Scetbon et al. '21):
>
> * _"A key observation when entropy is added to the coupling is that the more entropy is added, the lower the rank."_ (29);
> * _"a useful parallel can be drawn between (LOT) and that of the vanilla Sinkhorn algorithm, in the sense that they propose different regularization schemes. Indeed, the (discrete) path of solutions obtained by (LOT) when varying $r$ between $1$ and $\min(n,m)$ can be seen as an alternative to the entropic regularization path. Both paths contain at their extremes the original OT solution (maximal rank and minimal entropy) and the product of marginals (minimal rank and maximal entropy)."_ (30)
>
> Now because of this correspondence of small epsilon with large rank, and large epsilon with low-rank, the analogue of annealing in the parameter $\epsilon$ in entropy-regularized OT, i.e. gradually decreasing $\epsilon \to 0$ according to some schedule, is to initialize at a low-rank plan, and then to gradually increase the rank from low to full.
>
> In our case, this gradual rank increase is accomplished implicitly. At each scale $t = 1, \dots, \kappa$ this plan $\mathbf{P}^t$ is made explicit in our supplement, equation (S8). Our rank-annealing schedule $(r_1, \dots, r_\kappa)$ describes the sequence of multiplicative factors by which the rank of this explicit plan will increase at each successive scale. The partial products of these, denoted by
> $(\rho_1, \dots, \rho_\kappa)$, are the ranks of the plans $\mathbf{P}^1, \dots, \mathbf{P}^\kappa$. We will clarify this relationship in future versions, and distinguish the notion of a rank-annealing schedule from the secondary discussion of how to efficiently choose such a schedule under given memory constraints.

---

### Decision · Program_Chairs · 2025-05-01

**Decision:**

Accept (oral)

**Comment:**

The paper proposes a scalable and elegant framework for computing full-rank optimal transport using a hierarchical refinement of low-rank solutions. The method combines insights from low-rank OT and multiscale partitioning to construct bijective transport plans while maintaining linear space complexity and favorable runtime scaling. Reviewers consistently noted the novelty of this approach, the solid theoretical grounding—including formal propositions—and the wide range of convincing experiments across synthetic, biological, and large-scale visual datasets. The rebuttal successfully addressed concerns about missing comparisons and limitations in the discussion of related literature. In particular, the authors added benchmarks against mini-batch OT, provided further clarity on implementation details, and drew connections to hierarchical matrix methods. They also committed to broadening the experimental coverage and referencing additional hierarchical OT and domain adaptation works in the final version. These responses were well-received by the reviewers. Given the conceptual clarity, practical impact, and strong reviewer support, I recommend acceptance.